# ERLIN1/2 scaffolds bridge TMUB1 and RNF170 and restrict cholesterol esterification to regulate the secretory pathway

Matteo Veronese[1,2], Sebastian Kallabis[1,2,*], Alexander Tobias Kaczmarek[1,2,*], Anushka Das[1,2], Lennart Robers[1,2], Simon Schumacher[2,3], Alessia Lofrano[1,2], Susanne Brodesser[2,3], Stefan Müller[2], Kay Hofmann[1], Marcus Krüger[1,2,4], Elena I Rugarli[1,2,4]

**Complexes of ERLIN1 and ERLIN2 (ER lipid raft–associated 1 and 2) form large ring-like cup-shaped structures on the endoplasmic reticulum (ER) membrane and serve as platforms to bind cholesterol and E3 ubiquitin ligases, potentially defining functional nanodomains. Here, we show that ERLIN scaffolds mediate the interaction between the full-length isoform of TMUB1 (transmembrane and ubiquitin-like domain–containing 1) and RNF170 (RING finger protein 170). We identify a luminal N-terminal conserved region in TMUB1 and RNF170, which is required for this interaction. Three-dimensional modelling shows that this conserved motif binds the stomatin/prohibitin/flotillin/HflKC domain of two adjacent ERLIN subunits at different interfaces. Protein variants that preclude these interactions have been previously linked to hereditary spastic paraplegia. Using omics-based approaches in combination with phenotypic characterization of HeLa cells lacking both ERLINs, we demonstrate a role of ERLIN scaffolds in limiting cholesterol esterification, thereby favouring cholesterol transport from the ER to the Golgi apparatus and regulating Golgi morphology and the secretory pathway.**

## Introduction

The ER is the largest membrane-bound organelle and is involved in essential functions, including protein and lipid biosynthesis and calcium homeostasis. Similar to other cellular membranes, the ER lipid bilayer is organized in protein and lipid nanodomains, which define its major regions (nuclear envelope, cisternae, and peripheral tubules), compartmentalize specific processes, and mediate dynamic interactions with other cellular organelles at membrane contact sites (English & Voeltz, 2013). E3 ubiquitin ligases embedded in the ER membrane regulate the turnover of ER-resident proteins and remove misfolded secretory cargos during biogenesis in a process called ER-associated degradation (ERAD), thus preventing proteotoxicity (Christianson & Carvalho, 2022). How these regulatory enzymes are compartmentalized in the ER membrane is poorly understood.

The homologous ERLIN1 and ERLIN2 are ER-localized members of the evolutionarily conserved stomatin/prohibitin/flotillin/HflKC (SPFH) family of proteins, characterized by a conserved module of ~180–200 amino acids, which functions to scaffold lipids and proteins (Browman et al, 2007). Topological studies of ERLIN2 have shown that it is a type II ER membrane protein, containing a four–amino-acid-long cytosolic tail, followed by a transmembrane domain, the SPFH domain, and a coiled-coil domain (Browman et al, 2007). Like other members of the SPFH family, ERLIN1 and ERLIN2 assemble in a large ring-shaped hetero-oligomeric complex, likely formed by 24 subunits (Qiao et al, 2022; Yokoyama & Matsui, 2023). ERLIN1 and ERLIN2 have been first isolated as components of cholesterol-rich domains of the ER (Browman et al, 2006). Subsequent studies have established that ERLINs can directly bind cholesterol (Huber et al, 2013; Hulce et al, 2013). Moreover, acute down-regulation of ERLIN1 and ERLIN2 leads to the canonical activation of sterol regulatory element–binding proteins (SREBPs) and their target genes, leading to accumulation of lipid droplets (LDs) (Huber et al, 2013). As an underlying mechanism, a role of ERLINs in binding and stabilizing INSIG1 has been proposed (Huber et al, 2013).

Several pieces of evidence link ERLIN1 and ERLIN2 to ERAD, via binding and regulating membrane-embedded E3 ubiquitin ligases.

---

[1]Institute for Genetics, University of Cologne, Cologne, Germany   [2]Cologne Excellence Cluster on Cellular Stress Responses in Aging-Associated Diseases (CECAD), Cologne, Germany   [3]Faculty of Medicine and University Hospital Cologne, Cologne, Germany   [4]Center for Molecular Medicine (CMMC), University of Cologne, Cologne, Germany

Correspondence: Elena.rugarli@uni-koeln.de
Sebastian Kallabis's present address is Department of Systems Immunology and Proteomics, Institute of Innate Immunity, Medical Faculty, University of Bonn, Bonn, Germany
*Sebastian Kallabis and Alexander Tobias Kaczmarek contributed equally to this work

ERLINs have been found in association with the autocrine motility factor receptor (AMFR) (Jo et al, 2011), RNF170 (Pearce et al, 2007; Lu et al, 2011), RNF185 (Fenech et al, 2020; van de Weijer et al, 2020), and RNF5 (Fenech et al, 2020), although the exact organization of these complexes has not been established yet. In concert with RNF170, ERLIN2 is involved in the ERAD of the inositol 1,4,5-triphosphate (IP$_3$) receptors and other model ERAD substrates. ERLIN2 associates with the IP$_3$ receptors shortly after their activation and recruits the RING domain–containing protein RNF170 to ubiquitinate them (Pearce et al, 2007; Lu et al, 2011). ERLIN2 and AMFR were implicated in the sterol-accelerated degradation of 3-hydroxy-3-methylglutaryl co-enzyme reductase (HMGR), a rate-limiting enzyme for cholesterol synthesis (Jo et al, 2011). Moreover, a role of ERLIN2 in regulating the ubiquitination of the WNT secretory factor EVI was recently determined (Wolf et al, 2021). Finally, ERLINs compartmentalize the intramembrane rhomboid–related protease RHBDL4 to control proteolysis of aggregation-prone luminal ERAD substrates (Bock et al, 2022). Altogether, these data have fostered the hypothesis that ERLINs define ER cholesterol-enriched nanodomains involved in different ERAD and proteolytic pathways.

Elucidating how ERLIN complexes interact with different E3 ubiquitin ligases and determining the cellular processes controlled by the ERLIN complexes are crucial, because mutations in *ERLIN1* or *ERLIN2* gene have been associated with motoneuronal diseases, such as hereditary spastic paraplegia (HSP), primary lateral sclerosis, and amyotrophic lateral sclerosis (Alazami et al, 2011; Yildirim et al, 2011; Al-Saif et al, 2012; Wakil et al, 2013; Rydning et al, 2018; Tunca et al, 2018; Park et al, 2020; Srivastava et al, 2020; Kume et al, 2021; Qiao et al, 2022).

Here, we show that complexes of ERLIN1 and ERLIN2 limit cholesterol esterification, thereby promoting cholesterol trafficking from the ER to the Golgi apparatus and regulating the secretory pathway. In addition, we demonstrate that ERLIN scaffolds mediate the interaction between RNF170 and a long isoform of TMUB1 (TMUB1-L). TMUB1-L and RNF170 bind ERLIN monomers via a conserved domain, and the interaction interface in ERLIN1 and ERLIN2 is targeted by pathogenic mutations in HSP. Our data define the ERLIN complex as a crucial scaffold that couples cholesterol homeostasis to regulatory ERAD pathways controlled by RNF170 and TMUB1.

## Results

### The ERLIN complex mediates the interaction between TMUB1 and RNF170

To shed light on the processes regulated by complexes of ERLIN1 and ERLIN2, we introduced out-of-frame deletions in exon 1 of *ERLIN1* and exon 2 of *ERLIN2* using CRISPR/Cas9 gene editing in HeLa cells. Double knock-out (DKO) cells showed reduced mRNA levels of *ERLIN1* and *ERLIN2* consistent with RNA nonsense-mediated decay (Fig 1A), and lack of protein expression by Western blot using antibodies that recognize regions downstream of the deletions (Fig 1B). To control for off-target effects of gene editing, we used retroviral transduction in DKO cells to re-express the

proteins. For further experiments, we selected a clone that showed near-to-endogenous levels of ERLIN2 and the substantial re-expression of ERLIN1 (DKO$^{+E1/E2}$) (Fig 1B). Immunofluorescence using a specific antibody against ERLIN2 showed a staining consistent with localization to the ER that was lost in the DKO and restored in DKO$^{+E1/E2}$ cells (Fig 1C). Furthermore, recovery of detergent-resistant membranes (DRMs) after solubilization with a non-ionic detergent (Triton X-100) by flotation using an OptiPrep gradient revealed accumulation of both endogenous and re-expressed ERLIN1 and ERLIN2 in the same fraction as other DRM markers, such as flotillin-1 (FLOT1) and KIDINS220 (Fig 1D). This result agrees with the known accumulation of ERLINs in cholesterol-rich lipid nanodomains (Browman et al, 2006).

Currently, it is unclear whether different ERLIN complexes exist in association with distinct E3 ubiquitin ligases or whether ERLINs organize membrane domains containing more than one E3 ubiquitin ligase. A limitation of earlier studies is that they have been performed in overexpression conditions, which may promote unspecific interactions. We employed anti-ERLIN2 antibodies to pull down the endogenous complex and identify interacting proteins by label-free quantitative mass spectrometry (MS) (Fig 1E and Table S1). Among the most enriched proteins in the immunoprecipitation (IP) of WT cells were the molecular partner ERLIN1 and the E3 ubiquitin ligase RNF170. In addition, two other ERAD regulators, TMEM259 (transmembrane protein 259, also known as membralin) and TMUB1, were highly and significantly enriched, in agreement with previous studies (Fig 1E) (Jo et al, 2011; van de Weijer et al, 2020). We also identified the trans-2,3-enoyl-CoA reductase (TECR), an endoplasmic reticulum multipass protein that is involved in the fatty acid elongation cycle. In addition, ERLIN2 IP included other proteins that associate with DRMs, such as FLOT1 and stomatin (STOM), or proteins that traffic along the lipid rafts, as revealed by pathway analysis (Fig 1F). We also detected proteins belonging to other organelles, such as the mitochondrial voltage-dependent anion-selective channel protein 2 (VDAC2), the peroxisomal ATP–binding cassette subfamily D member 3 (ABCD3), and the small GTPases ARF1 and ARF4 (ADP-ribosylation factor 1 and 4), which are involved in vesicular trafficking (Adarska et al, 2021), possibly reflecting the presence of ERLINs at the contact sites between the ER and other organelles.

TMUB1 contains a ubiquitin-like (UBL) domain, and acts as an escortase stabilizing intermediates of membrane proteins during retro-translocation and delivering them to p97/VCP (valosin-containing protein) (Wang et al, 2022). A long and a short isoform of TMUB1 (from now on TMUB1-L and TMUB1-S) deriving from alternative initiation of translation have been previously reported (Fig 2A) (Castelli et al, 2014; van de Weijer et al, 2020; Wang et al, 2022). Reciprocal co-IP experiments demonstrated that ERLIN2 interacts specifically with TMUB1-L (Fig 2B and C). Furthermore, the amount of TMUB1-L recovered in the DRM fraction was decreased in DKO cells and rescued upon the re-expression of the ERLINs (Fig 2D–F), suggesting a role of ERLIN complexes to cluster TMUB1-L in specific membrane nanodomains. In contrast, the distribution of TMUB1-S and of other ERLIN2 interactors remained unaffected (Figs 2D–F and S1). Unfortunately, we could not test whether the lack of ERLINs impairs targeting of RNF170 to the DRMs because of a lack of suitable antibodies.

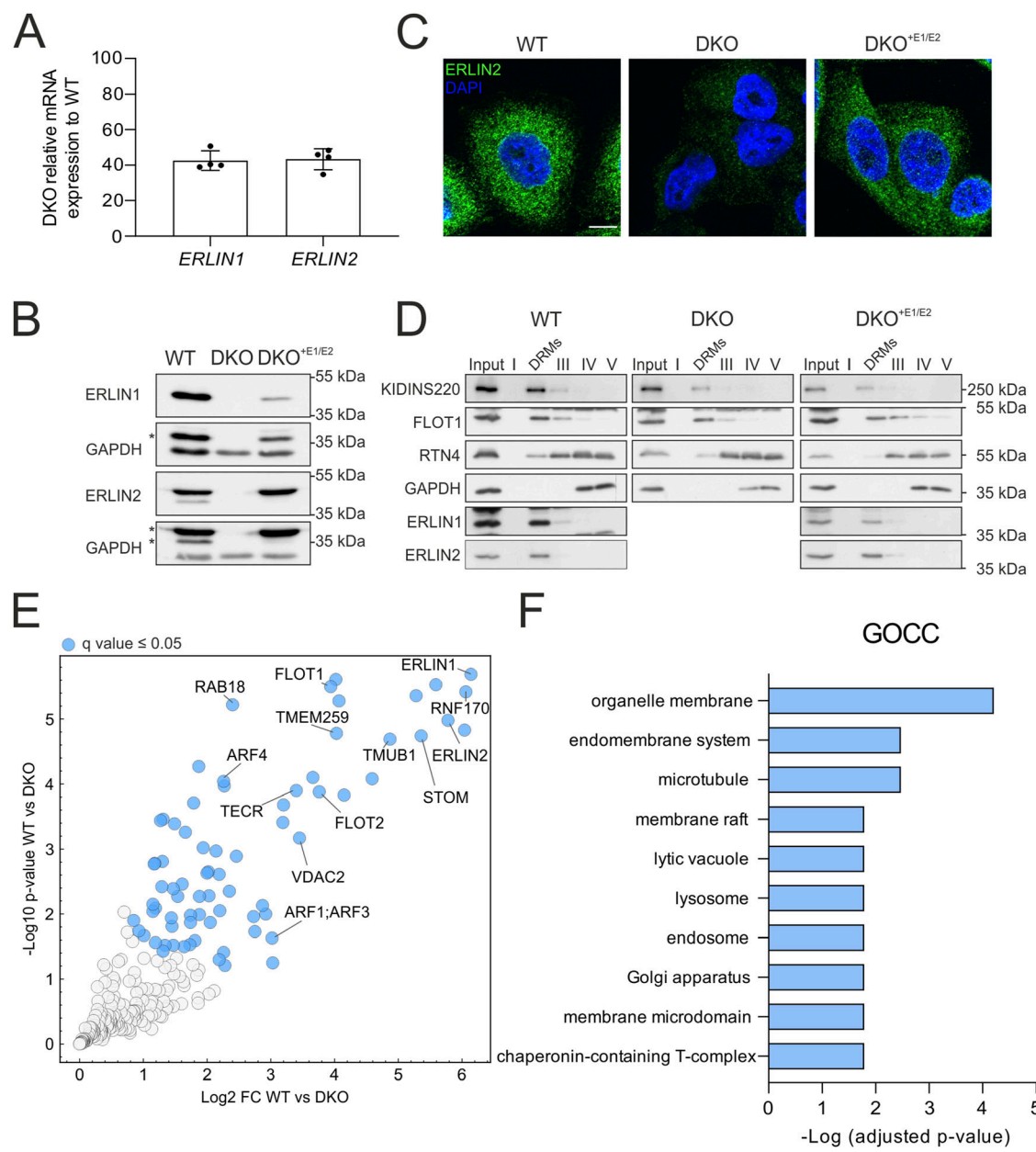

**Figure 1.  Generation of cells lacking ERLIN1 and ERLIN2 and identification of the interactome of ERLIN2.**
**(A)** Expression levels of *ERLIN1* and *ERLIN2* mRNAs measured by qRT–PCR normalized for *GAPDH* in double knock-out (DKO) compared with WT (set to 100). **(B)** Western blot for ERLIN1 and ERLIN2 in WT, DKO, and DKO[+E1/E2]. GAPDH was used as a loading control. Asterisks mark unspecific bands. **(C)** Immunofluorescence staining of ERLIN2 in WT, DKO, and DKO[+E1/E2]. **(D)** Western blot for ERLIN1 and ERLIN2 in the input and different fractions of detergent-resistant membrane (DRM)–isolating gradients from WT, DKO, and DKO[+E1/E2] cells. FLOT1 and KIDINS220 are DRM markers, RTN4 is a non-DRM protein, and GAPDH is a soluble protein. **(E)** Volcano plot of endogenous ERLIN2 interactors in WT compared with DKO cells. N = 4 biological replicates. Significantly enriched proteins (*Q*-value ≤ 0.05) in DKO are labelled in blue. **(F)** Enriched gene ontology cellular component (GOCC) terms of proteins with *Q*-value ≤ 0.05 performed using the gProfiler webtool; all proteins identified in the analysis were used as background.
Source data are available for this figure.

To gain further insights into the specific function of TMUB1-L, we generated TMUB1-L–specific KO cell lines using an inactive Cas9 fused with an adenine deaminase to mutagenize the first ATG (Gaudelli et al, 2017) (Fig S2A), and used one of the clones as control in IP experiments, followed by MS (Table S2). ERLIN1 and ERLIN2 emerged as the top proteins co-immunoprecipitated by TMUB1-L,

validating our earlier findings (Fig 2G). Moreover, we confirmed the interaction of TMUB1-L with RNF170, TMEM259, RNF185, FAF2 (FAS-associated factor 2), and VCP (Wang et al, 2022). FAF2 is anchored to the ER by a hairpin domain, contains both a UBA and UBX domain, and has been previously implicated in ERAD pathways (Mueller et al, 2008; Olzmann et al, 2013). Intriguingly, AlphaFold-Multimer predicts not only

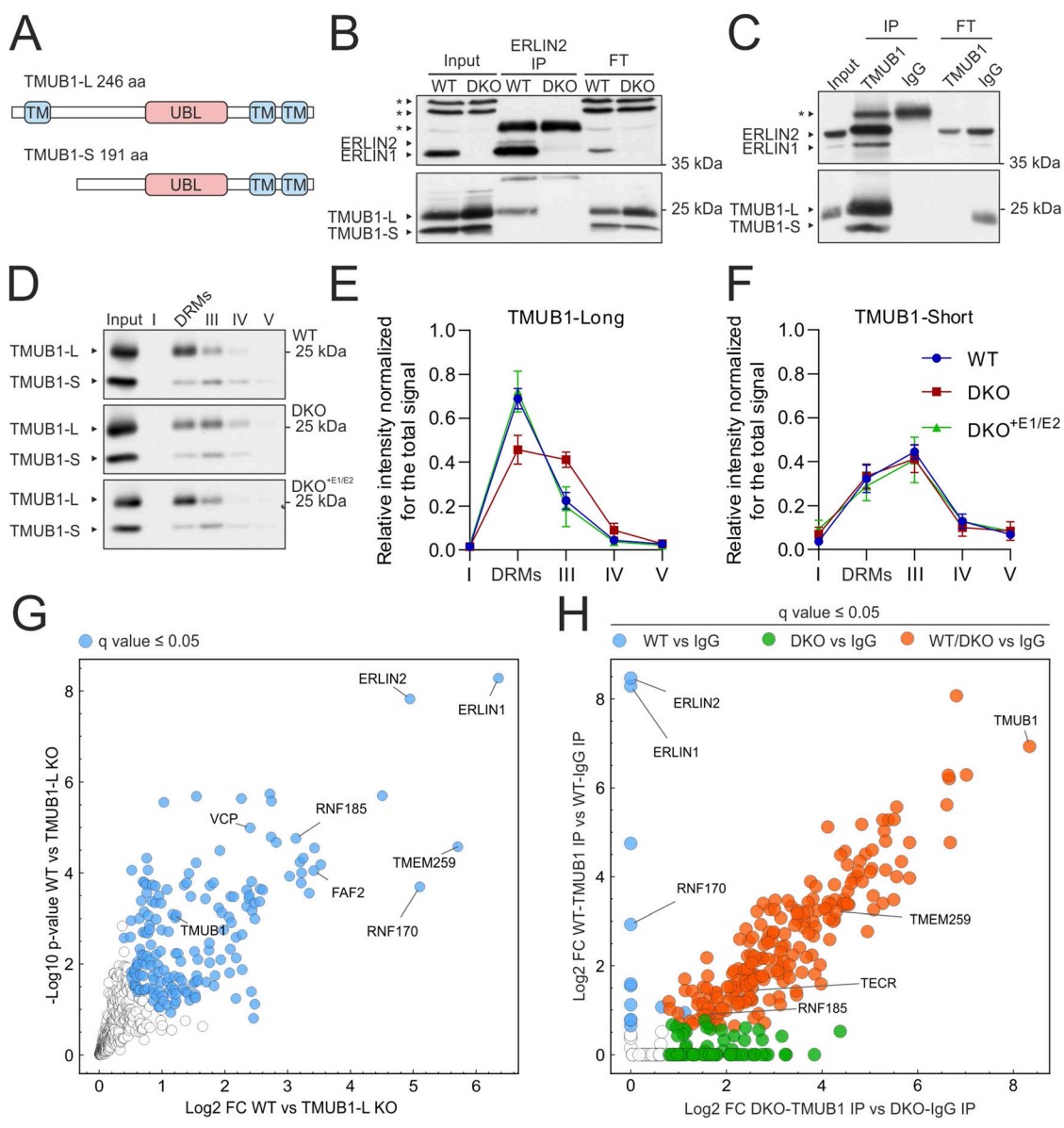

**Figure 2. ERLINs interact with TMUB1-L and mediate the interaction of TMUB1-L with RNF170.**
**(A)** Scheme of the domain structure of the two human TMUB1 isoforms. **(B, C)** Western blots of reciprocal co-IPs of endogenous ERLIN2 and TMUB1 in HeLa cells. Input is equal to 10% of the proteins used for the IP. Asterisks mark unspecific bands. FT, flow-through. **(D)** Western blot of TMUB1 in the input and different fractions of detergent-resistant membrane–isolating (DRM) gradients in WT, double knock-out (DKO), and DKO$^{+E1/E2}$. **(E, F)** Quantification of TMUB1-L (E) and TMUB1-S (F) in the different fractions. N = 4 biological replicates. **(G)** Volcano plot of endogenous TMUB1 interactors in WT compared with TMUB1-L KO cells. N = 4 biological replicates. Significantly enriched proteins ($Q$-value ≤ 0.05) in WT are labelled in blue. **(H)** Correlation of protein fold changes of a TMUB1 IP in WT versus ERLIN DKO cells. Proteins significantly enriched ($Q$-value ≤ 0.05) only in WT cells are labelled in blue, and only in DKO in green, whereas proteins immunoprecipitated in both conditions are depicted in orange. Source data are available for this figure.

the known interaction of the UBX domain with VCP (Schuberth & Buchberger, 2008), but also a possible interaction between the UBA domain of FAF2 and the UBL domain of TMUB1-L (Fig S2B–D).

We then asked whether any of the molecular interactions of TMUB1-L depend on the presence of ERLIN scaffolds. To this purpose, we used antibodies detecting both TMUB1 isoforms for IP experiments in WT and DKO cells (Figs 2H and S2E and Table S2). Many proteins enriched in the TMUB1 IP belonged to a nucleolar

compartment but were likely due to a cross-reactivity of the antibody, because a nucleolar staining could not be suppressed by down-regulation of TMUB1 with siRNA (Fig S2F and G). Surprisingly, lack of ERLINs completely prevented the interaction of TMUB1 with RNF170 (Fig 2H), whereas the TMUB1-RNF185-TMEM259 interaction previously reported was still detected in DKO cells. We conclude that the ERLIN scaffolds are required to cluster TMUB1-L and RNF170 in ER membrane nanodomains.

## A conserved luminal domain in TMUB1-L and RNF170 interacts with the ERLIN complex

TMUB1-L and RNF170 have a similar topology with a luminal domain, three transmembrane domains, and a cytosolic UBL and RING domain, respectively (Fig 3A). Inspection of the N-terminal luminal domain of TMUB1-L, which is required for interaction with ERLINs, revealed the presence of a stretch of seven amino acids that was conserved in the N-terminal domain of RNF170 (Fig 3B). We used AlphaFold-Multimer (Evans et al, 2021 Preprint), to model the three-dimensional interaction between TMUB1-L, RNF170, and a dimer of ERLIN1 and ERLIN2 (Fig 3C). The best model predicted that the conserved N-terminal motif of RNF170 and TMUB1 interacts with the SPFH domain of each ERLIN subunit at different interfaces. The first interaction occurs with a $\beta$-sheet in the ERLINs. Hydrogen bonds are predicted between Tyr51 of ERLIN1 and Gly21 of RNF170, whereas Phe49 and Leu51 in ERLIN2 contact Gly6 and Gly10 of TMUB1, respectively. On the second interface, Glu5 of TMUB1-L or the corresponding Glu20 of RNF170 binds to the adjacent ERLIN subunit with hydrogen bonds predicted with Arg38 and Thr44 in ERLIN1 and Arg36 and Thr42 in ERLIN2 (Fig 3D).

A Gly50Val substitution in ERLIN1 has been implicated in autosomal recessive SPG62 (Novarino et al, 2014), whereas His50Tyr (ClinVar ID: 947803) and Arg36Lys (ClinVar ID: 1720746) in ERLIN2 have been reported as a variant of unknown significance in individuals with spastic paraplegia. We explored whether these variants alter the interaction of the ERLINs with TMUB1-L and RNF170, by modelling the interaction after mutating the corresponding residues in ERLIN2 (Fig 3E). The size of the ERLIN2 $\beta$-sheet between Gly48 and His50 is shortened by introducing the mutations, and the hydrogen bonds predicted between TMUB1-L and the ERLIN subunits at the two interaction interfaces are lost (Fig 3E). The expression of ERLIN2-FLAG mutant constructs carrying these variants individually or in combination showed that the Gly48 is required for the stability of the protein (Fig 3F). Similarly, the simultaneous mutation of the two residues predicted to form hydrogen bonds with both TMUB1-L and RNF170 also induces protein degradation (Fig 3F). These data support the hypothesis that these variants are pathogenic and that the interaction of ERLINs with TMUB1-L and RNF170 plays a role in stabilizing the ERLIN complex.

## Loss of ERLINs perturbs pathways linked to cell adhesion and vesicular trafficking

To uncover pathways regulated by the ERLIN complex, we first used unbiased omics-based approaches. Transcriptomics analysis of WT and DKO cells confirmed activation of the SREBP pathway upon ERLIN depletion (Huber et al, 2013); however, this was milder than previously reported (Fig 4A and Table S3). Over-representation analysis of the DKO transcriptome showed a significant down-regulation of GO terms related not only to the ER, but also to focal adhesion, cell adhesion, and cadherin binding (Fig 4B). We used label-free quantitative MS to reveal changes at the proteome level in DKO cells. We only considered proteins measured by at least two peptides, showing significant changes in abundance between DKO

and WT, and substantially rescued by re-expressing the ERLINs. Of 2,742 proteins measured in the post-nuclear supernatant, only 34 fulfilled these criteria (Fig 4C and Table S4). Many of these proteins functionally belonged to pathways related to sterol and lipid metabolism and cell adhesion (Fig 4C). Most of these changes were mirrored by transcriptional alterations (Fig 4C). ER-resident enzymes involved in neutral lipid and cholesterol metabolism (ACSL1, HMGCR, and SQLE) and regulated by SREBPs were increased. Furthermore, we found that $\beta$-catenin (CTNNB1) was increased in DKO cells and rescued in the DKO$^{+E1/E2}$ cells. Alteration of the $\beta$-catenin pathway was further confirmed via Western blot (Fig S3A and B) and was paralleled by a migration defect in a scratch assay, in which DKO cells took more time to fill a gap on a plate compared with WT and DKO$^{+E1/E2}$ cells (Fig S3C and D).

We reasoned that the lack of the scaffolding function of the ERLIN complex in the ER may not necessarily affect overall protein abundance but may change their dynamic subcellular distribution by impairing recruitment to cholesterol-enriched nanodomains. We therefore combined the flotation gradient used to separate DRMs with stable isotope labelling by amino acids in cell culture (SILAC) (Ong et al, 2002) and analysed protein abundance in five collected fractions with different densities. The major advantage of this approach is allowing us to combine WT, DKO, and DKO$^{+E1/E2}$ cell lysates before loading them on the flotation column, thus reducing the variability of fraction collection among samples (Fig 5A). Proteins associated with DRMs are recovered in low-density fractions; however, they can shift from one fraction to another upon altered cellular lipid composition or signalling stimulation (Simons & Toomre, 2000; Foster et al, 2003). This approach confirmed that in WT cells, ERLIN1, ERLIN2, and other DRM-associated proteins, such as FLOT1 and FLOT2, are enriched in fraction II, as previously detected by Western blot (Figs S4 and 1D). The overall distribution of the $\log_2$ normalized protein ratio was similar between all comparisons in fractions II to V (Fig 5B). Fraction I contained significantly less proteins than the other fractions, which accumulated in the DKO compared with the other genotypes (Fig 5B). We used unsupervised clustering to identify proteins that were significantly changed in the three comparisons (KO/WT, DKO$^{+E1/E2}$/DKO, and DKO$^{+E1/E2}$/WT). We identified cluster 170 in fraction I and cluster 155 in fraction II as containing proteins that were up-regulated in the DKO and rescued in the DKO$^{+E1/E2}$ cells (Fig 5C, D, F, and G and Table S5). Pathway analysis showed that these clusters were enriched in proteins belonging to adherens and anchoring junctions, to the plasma membrane, and to membrane-bound vesicles (Fig 5E and H). Surprisingly, cluster 577 in fraction IV contained proteins belonging to the same GO terms, showing, however, an opposite trend compared with the previous two clusters. Thus, proteomics of DKO cells suggests perturbations of proteins that follow the secretory route.

## Loss of the ERLIN complex leads to a collapse of ER tubules and Golgi fragmentation

Based on the previous results, we hypothesized that trafficking along the secretory compartment may be impaired in DKO cells. As a first step, we used antibodies against reticulon 4 (RTN4), which is enriched in the tubular ER, to assess whether the general morphology of the ER was affected in DKO cells. Although the total levels of RTN4 did not change in cells of different genotypes (Table S4), immunofluorescence

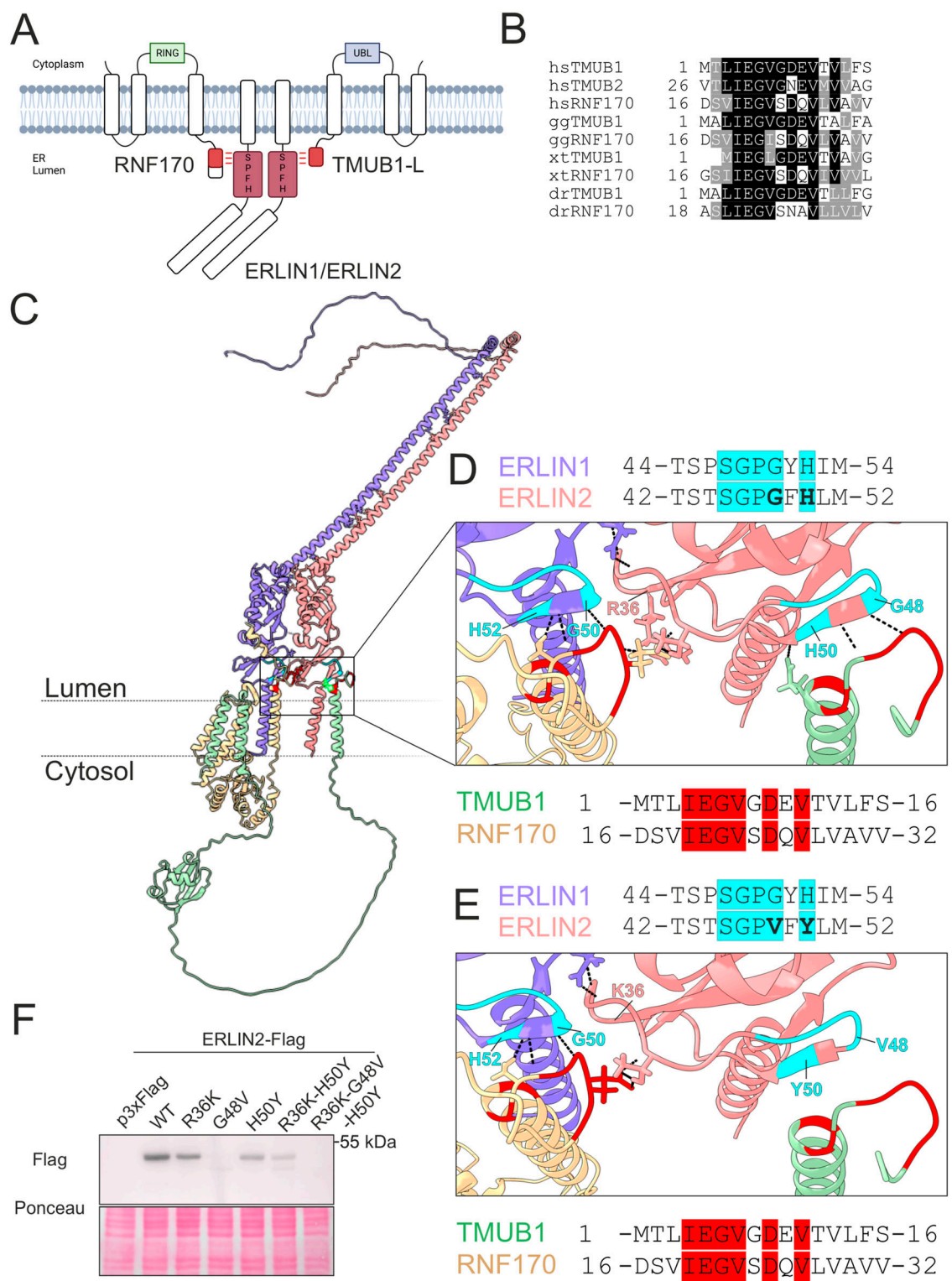

**Figure 3. Model of the ERLIN1/2-RNF170-TMUB1-L complex.**
**(A)** Schematic representation of the ERLIN1/2-RNF170-TMUB1-L complex topology. **(B)** N-terminal conserved motif alignment in TMUB1-L, TMUB2, and RNF170. **(C)** AlphaFold-Multimer model of ERLIN1/2-TMUB1-L-RNF170. ERLIN1 is represented in violet, ERLIN2 in pink, TMUB1 in green, and RNF170 in yellow. **(D)** Enlargement of the docking site with predicted H-bonds. Conserved sequences between ERLIN1 and ERLIN2 are shown in cyan and between TMUB1 and RNF170 in red. **(E)** Enlargement of the same complex containing the ERLIN2 R36K-G48V-H50Y variants. **(F)** Western blot of transfected ERLIN2-FLAG variants expressed in HeLa cells detected with anti-FLAG antibody.
Source data are available for this figure.

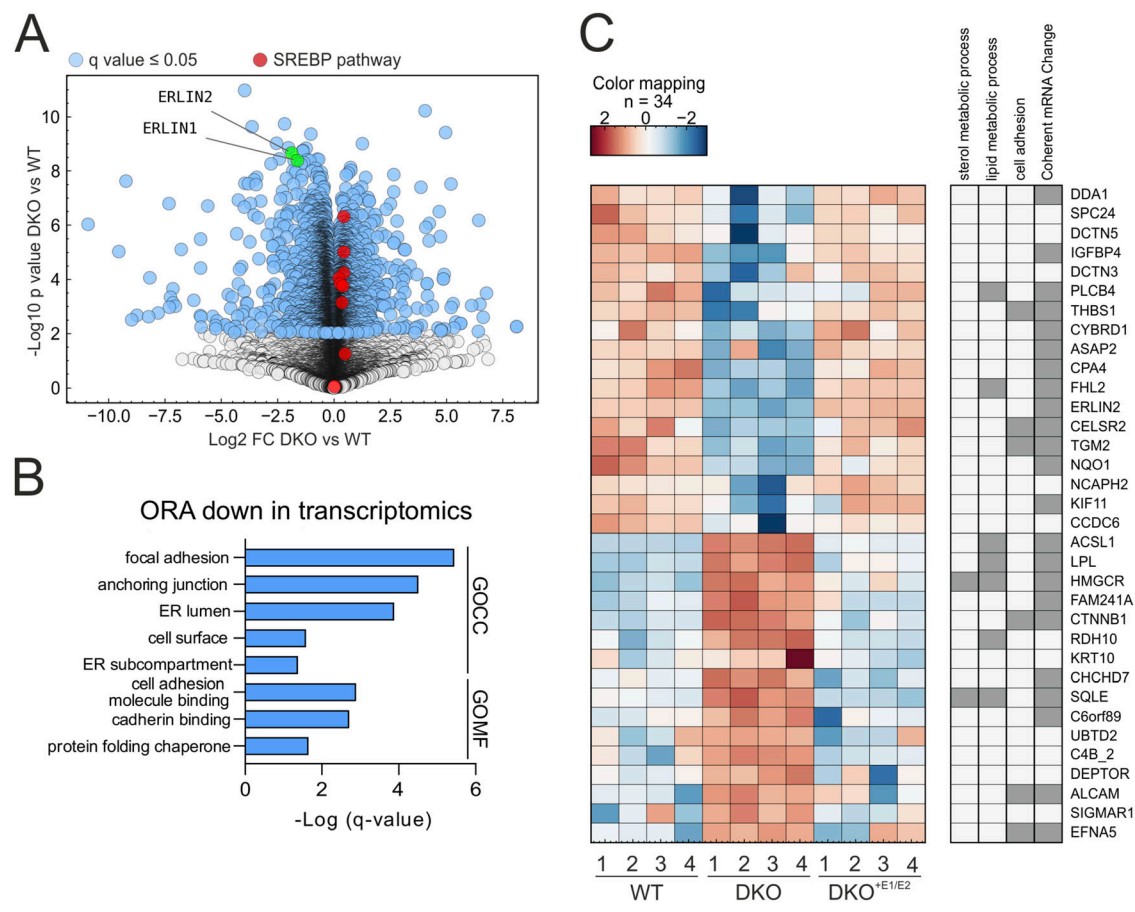

**Figure 4. RNA-seq and proteomics of double knock-out (DKO) reveal perturbed pathways.**
**(A)** Volcano plot of the RNA-seq analysis. Genes with $Q$-value ≤ 0.05 are labelled in blue, and known SREBP target genes are labelled in red. **(B)** Over-representation analysis of the down-regulated genes from the RNA-seq. N = 4 biological replicates. GOCC, gene ontology cellular component; GOMF, gene ontology molecular function. **(C)** Heatmap of z-score normalized intensities of significant proteins in ANOVA ($Q$-value ≤ 0.05) with $\log_2$ fold change ≥ 0.59 or ≤ −0.59 in WT versus DKO, and DKO versus DKO$^{+E1/E2}$, but not in WT versus DKO$^{+E1/E2}$. N = 4 biological replicates.

analysis showed altered ER morphology in DKO cells, which were intriguingly depleted of ER peripheral tubules in comparison with both WT and DKO$^{+E1/E2}$ cells (Fig 6A and B). We also noticed that the cell area labelled by RTN4 staining in DKO cells appeared smaller 24 h after plating (Fig S5A), although the cell size was unchanged when assessed in cytofluorometry (Fig S5B). Therefore, the ER phenotype may be influenced by a defect of the cells in spreading on the plate, consistent with the cell adhesion defect.

To assess trafficking along the secretory system, we employed the retention using selective hooks (RUSH) system (Boncompain et al, 2012). This system comprises a hook fused to streptavidin, which is bound to the donor compartment (the ER), and a reporter protein fused to a streptavidin-binding domain. The addition of biotin to the medium synchronously competes with the streptavidin-binding domain, releasing the reporter protein from the hook and allowing its trafficking along the secretory pathway. We employed the vesicular stomatitis virus G (VSVG) glycoprotein expressed with a fluorescent EGFP tag as a reporter and monitored in live imaging its trafficking after biotin addition in WT, DKO, and DKO$^{+E1/E2}$ cells (Fig S5C and Video 1, Video 2, and Video 3). In WT and DKO$^{+E1/E2}$ cells, VSVG-GFP translocated from the ER

to the Golgi compartment within ~10 min, followed by trafficking to the exocytic compartment. In ERLIN-deficient cells, VSVG-GFP moved from the ER to a fragmented vesicular compartment within 10 min after adding biotin, and then remained largely confined within these structures. When we analysed colocalization of the VSVG signal with the Golgi marker GM130 in cells fixed at different time points after biotin addition, we confirmed that in DKO cells, the VSVG-EGFP remained elevated in the Golgi after 60 min from biotin addition, in contrast to WT and DKO$^{+E1/E2}$ cells (Fig 6C and D). Furthermore, the loss of ERLINs induced Golgi fragmentation (Fig 6E and F), a phenotype that was rescued in DKO$^{+E1/E2}$ cells (Fig 6E–G). We then examined whether Golgi proteins changed their abundance in the different fractions analysed by SILAC among genotypes (Table S5), as a further indication of the organelle dispersal in DKO cells. ARF1 was affected in clusters 170, 155, and 577 in fractions I, II, and IV, respectively. Additional Golgi proteins, including GORASP2 (GRASP55), were changed in abundance in cluster 577 of fraction IV (Table S5). The steady-state levels of GM130 (GOLGA2) were unaffected in the post-nuclear supernatant proteomics but were perturbed in the fraction III analysed by SILAC (Tables S4 and S5). Thus, the lack of the ERLIN complex does not impair ER-to-Golgi trafficking

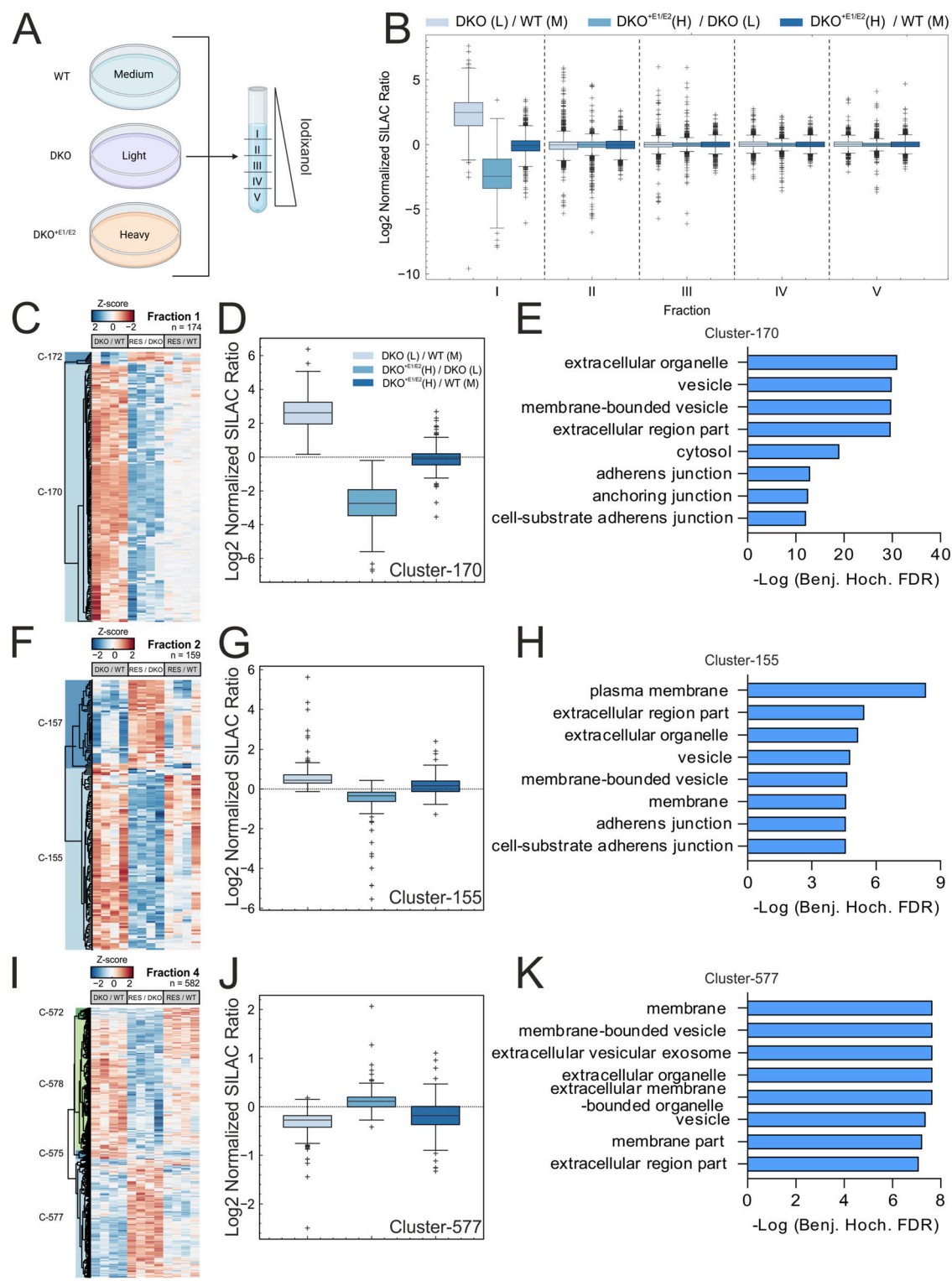

**Figure 5.   Lack of ERLINs alters the distribution of proteins in detergent-resistant membrane fractions.**
**(A)** Scheme of SILAC approach combined with the DRM isolation protocol. **(B)** Normalized SILAC ratio of each fraction. **(C, D, E, F, G, H, I, J, K)** Unsupervised cluster analysis of significantly regulated proteins (ANOVA $Q$-value ≤ 0.05) detected in fractions I, II, and IV reveals clusters of proteins that are perturbed in double knock-out and rescued in DKO$^{+E1/E2}$ (RES). Heatmaps of fractions I, II, and IV, respectively (C, F, I). Log$_2$ normalized SILAC ratio of clusters 170, 155, and 577 (D, G, J) and corresponding significantly enriched GO terms ($Q$-value ≤ 0.05) with Fisher's exact test (E, H, K).

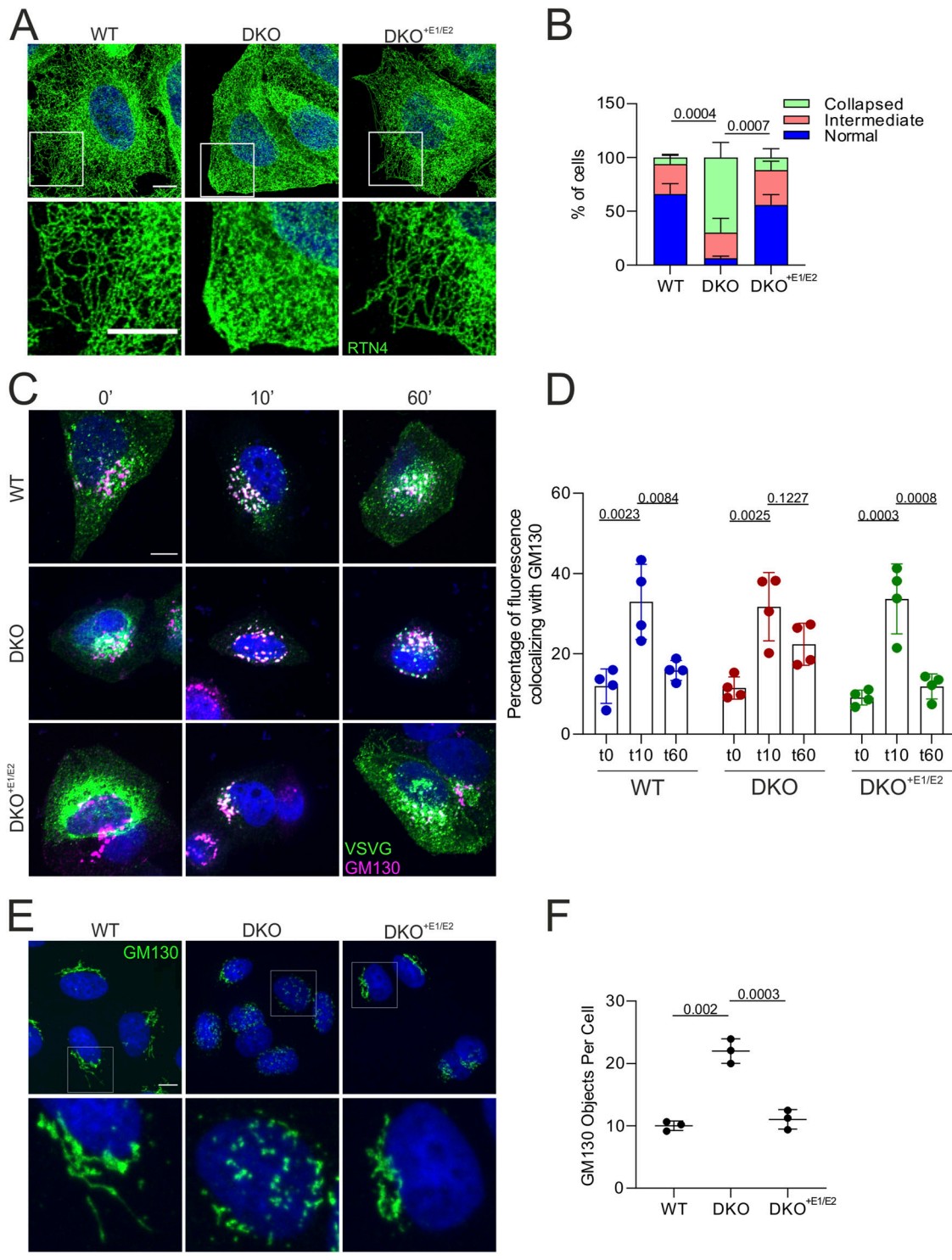

**Figure 6. Double knock-out (DKO) cells show defects along the secretory pathway.**
**(A)** Immunofluorescence of the ER using an anti-RTN4 antibody in cells of different genotypes. **(B)** For quantification of (A), ER tubules were classified as normal, intermediate, and collapsed. N = 3 independent experiments (at least 90 cells were analysed per genotype). SD is shown. *P*-value refers to the collapsed phenotype. **(C)** RUSH assay using the vesicular stomatitis virus G–GFP construct. Cells were fixed at indicated time points and stained for GM130 to label the Golgi apparatus. **(D)** Quantification of the percentage of GFP colocalizing with GM130 at each time point. The ratio between the GFP signal inside the GM130-positive area and the total GFP signal is shown. N = 4 biological replicates (at least 110 cells were analysed per genotype for each time point). **(E)** Immunofluorescence of the Golgi apparatus using an anti-GM130 antibody. **(F)** Quantification of the experiment as in (E). N = 3 biological replicates (at least 300 cells were analysed per genotype). Graphs show individual data points, mean, and SD. The statistical tests employed are one-way ANOVA with post hoc Tukey's multiple comparison test.

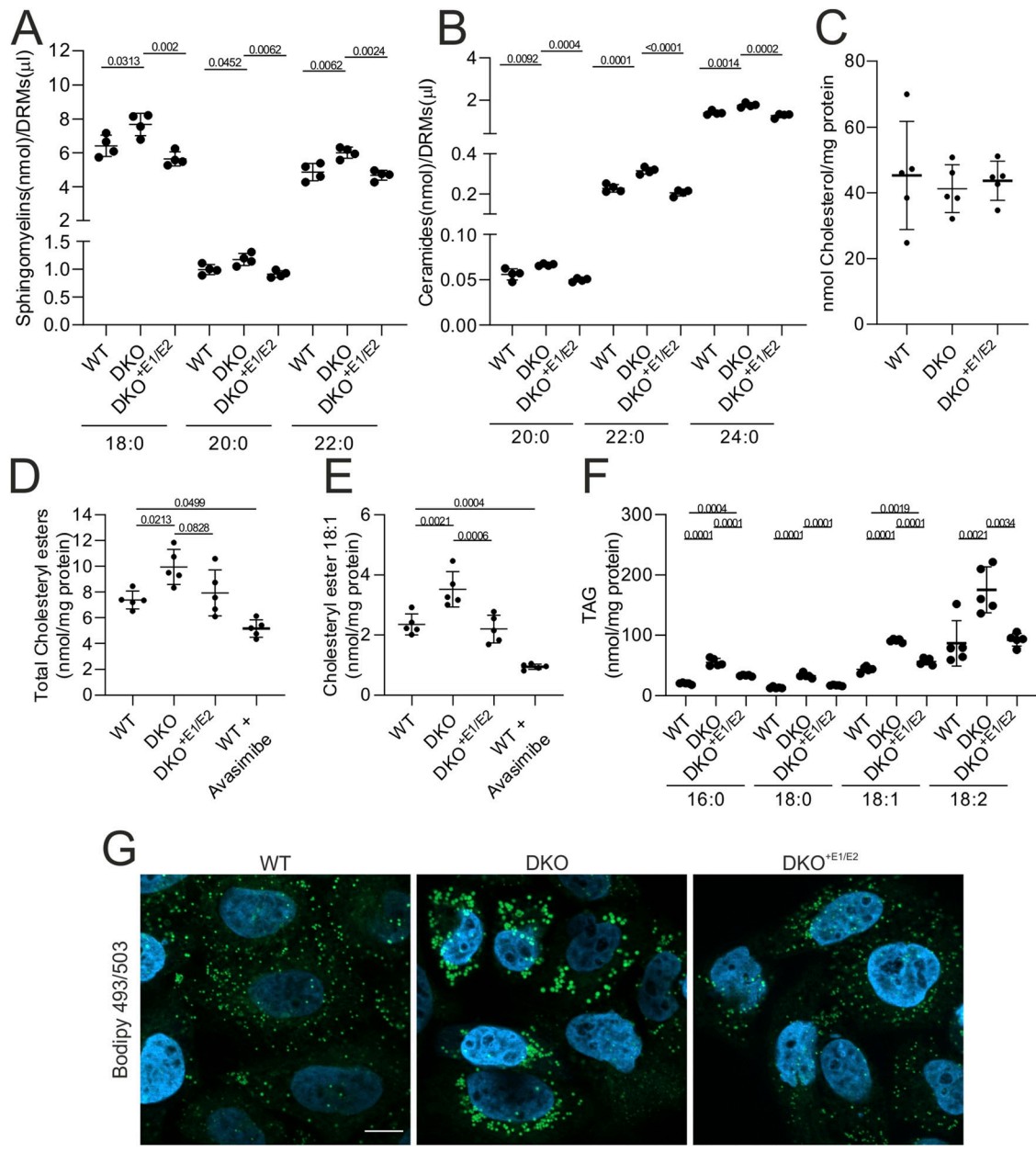

**Figure 7.   DKO cells show sphingolipid accumulation in detergent-resistant membranes and neutral lipid accumulation in LDs.**
**(A, B)** Sphingomyelins (A) and ceramides (B) accumulate in DRMs of DKO cells. N = 4 biological replicates. Only significant species are shown. **(C, D, E, F)** Lipidomics analysis of cholesterol (C), total cholesteryl ester (D), cholesteryl ester 18:1 (E), and TAGs (F). When indicated, WT cells were treated with avasimibe to confirm endogenous synthesis. N = 5 biological replicates. In (A, B, C, D, E, F), graphs show individual data points, mean, and SD. The statistical test shown is one-way ANOVA with post hoc Tukey's multiple comparison test. **(G)** Staining of LDs with Bodipy 493/503.

but induces morphological defects of the Golgi apparatus and affects post-Golgi trafficking. Notably, we observed dispersion of the Golgi apparatus also upon TMUB1-L depletion (Fig S5D and E).

### Excessive cholesterol esterification in the absence of ERLINs causes Golgi fragmentation

The Golgi apparatus is a central sorting station not only for proteins, but also for lipids that traffic along the secretory pathway. Although the presence of lipid rafts in the ER is still debated, it is accepted that

lipid nanodomains enriched with cholesterol and sphingolipids are present along the secretory pathway, with increasing abundance from the ER to the plasma membrane. These domains are implicated in the intracellular trafficking of proteins, such as GPI-linked receptors, integrins, and other adhesion molecules. By binding cholesterol at the ER, complexes of ERLINs may control the distribution of cholesterol in cellular membranes. We measured levels of ceramides and sphingomyelin in the DRMs of WT, DKO, and DKO[+E1/E2] cells using targeted lipidomics. We found a higher quantity of species of both lipids containing fatty acids with chain lengths between C18 and C24 in the

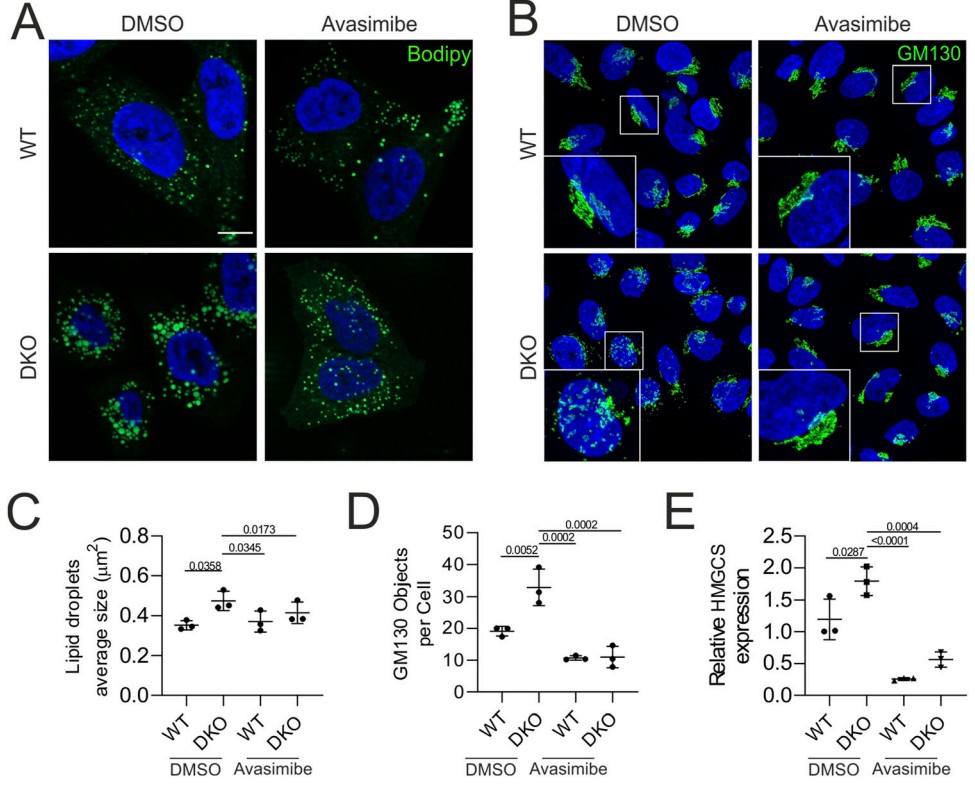

**Figure 8. Avasimibe treatment of DKO cells rescues LD accumulation, Golgi fragmentation, and *HMGCS* transcriptional up-regulation.**
**(A)** Bodipy 493/503 staining and **(B)** GM130 immunofluorescence of WT and DKO cells treated with and without avasimibe. **(C)** Quantification of LD size. N = 3 biological replicates (at least 130 cells were analysed per genotype). **(D)** Quantification of Golgi fragmentation. N = 3 biological replicates (at least 340 cells were analysed per genotype). **(E)** Relative *HMGCS* expression in WT and DKO cells with and without avasimibe. GAPDH was used as a control. In (C, D, E), graphs show individual data points, mean, and SD. The statistical test shown is the one-way ANOVA with post hoc Tukey's multiple comparison test.

DKO cells, which were restored to WT levels in the DKO$^{+E1/E2}$ cells (Fig 7A and B and Table S6). The protocol used for the preparation of DRMs prevented measuring cholesterol levels in these fractions. We therefore assessed the content of free cholesterol and cholesterol esters (CE) in the total cell lysates. Unesterified cholesterol was not affected upon ERLIN depletion (Fig 7C). However, total CE and especially the abundant CE 18:1 species were significantly increased when ERLINs were depleted and recovered to WT levels in DKO$^{+E1/E2}$ cells (Fig 7D and E and Table S6). Avasimibe, an inhibitor of SOAT1 (sterol O-acyl-transferase 1), the enzyme that catalyses the formation of fatty acid–cholesterol esters, strongly suppressed levels of CE18:1 in WT cells, confirming that this species is mainly synthesized endogenously (Fig 7E). Levels of triacylglycerides (TAGs) also increased in the DKO compared with both WT and DKO$^{+E1/E2}$ cells (Fig 7F and Table S6). Consistent with the accumulation of neutral lipids, larger LDs labelled by Bodipy 493/503 accumulated in DKO cells, a phenotype that was rescued upon re-expression of ERLINs (Fig 7G).

Esterification of excess cholesterol at the ER and its accumulation in LDs are observed when cholesterol transport from the ER to the Golgi is impaired (Mesmin et al, 2017). Cholesterol trafficking between these two organelles mostly occurs via non-vesicular transport, at sites of close apposition between the membranes of the organelles (Ikonen & Zhou, 2021). This transfer is mediated by oxysterol-binding proteins (OSBPs), which bind cholesterol at the ER and release it at the Golgi in exchange for a molecule of phosphatidylinositol-4-phosphate (Ikonen & Zhou, 2021). The FFAT (two phenylalanines—FF—in an acidic tract) domain of OSBP binds the protein VAPA (VAMP-associated protein A) at the ER, whereas

the pleckstrin homology (PH) domain binds ARF1 at the Golgi membrane. Notably, ARF1 was identified in the ERLIN2 interactome (Fig 1E), suggesting a close apposition between the ERLIN complex and the Golgi. Cholesterol balance in the Golgi is essential for protein transport, and both cholesterol depletion and cholesterol loading block trafficking of secretory proteins from the trans-Golgi network (Stüven et al, 2003). In addition, lack of cholesterol in the Golgi may lead to vesiculation (Stüven et al, 2003). DKO cells showed a modest but significant activation of SREBPs, indicating that they evade feedback inhibition by cholesterol synthesis. Although the destabilization of INSIG1 upon ERLIN knockdown could explain this result (Huber et al, 2013), an alternative explanation could be that increased cholesterol esterification keeps levels of cholesterol in the ER low, impairs transport to the Golgi, and prevents feedback inhibition of SREBPs. To test this hypothesis, we treated WT and DKO cells with avasimibe. Remarkably, inhibition of SOAT1 activity not only reduced the size of the LDs in DKO cells (Fig 8A and C), but also rescued the Golgi fragmentation (Fig 8B and D), supporting our hypothesis. SOAT1 inhibition suppressed the transcriptional up-regulation of 3-hydroxy-3-methylglutaryl-CoA synthase (HMGCS), a SREBP target gene in both WT and DKO cells, blunting the difference between genotypes (Fig 8E). We propose a model in which the ERLIN complex, by binding cholesterol, restricts its esterification and thus promotes its transport to the Golgi. The loss of ERLIN1/2 scaffolds on the one hand depletes the ER of cholesterol, activating the SREBP pathway, and on the other hand leads to Golgi vesiculation and defects in post-Golgi trafficking.

# Discussion

Our study identifies a key role of ERLIN1 and ERLIN2 scaffolds on the ER to mediate the interaction between TMUB1-L and RNF170 and to limit cholesterol esterification, thereby facilitating the transport of cholesterol to the Golgi. Lack of the ERLIN complex impairs the secretory pathway, leading to the collapse of the tubular ER, Golgi fragmentation, impaired post-Golgi trafficking, and cell migration defects. These results suggest novel pathogenic pathways underlying the degeneration of long axons of the corticospinal tracts in patients carrying mutations in *ERLIN1* or *ERLIN2*.

We show that the ERLIN complex functions as a platform to recruit the E3 ubiquitin ligase RNF170 and the long isoform of TMUB1. This interaction occurs via a conserved eight–amino-acid-long motif in the N-terminus of RNF170 and TMUB1-L, which can bind adjacent ERLIN subunits in a hetero-oligomeric complex, based on three-dimensional modelling using AlphaFold-Multimer. Consistently, lack of the N-terminal motif in TMUB1-S abrogates binding to the ERLINs, TMUB1 immunoprecipitates RNF170 in WT but not in DKO background, and less TMUB1-L is recovered in the DRM fraction in the DKO. The motif is unique for RNF170 and TMUB1 and is conserved in TMUB2, but not in other E3 ubiquitin ligases previously linked to the ERLINs. The interaction interface on the ERLIN side involves two luminal regions at the beginning of the SPFH domain. Particularly relevant is a β-sheet formed by a glycine, followed by an aromatic amino acid (Tyr in ERLIN1 and Phe in ERLIN2), and a histidine. Shortening of this β-sheet in ERLIN2 by mutating the glycine to valine to reproduce an HSP mutation found in *ERLIN1* (Novarino et al, 2014) results not only in the loss of the predicted hydrogen bond with TMUB1 and RNF170, but also in the destabilization of the protein, without impairing the protein oligomerization. A similar effect was obtained when concomitantly mutating the two ERLIN2-binding interfaces, by introducing mutations reported as a variant of unknown significance in patients with HSP. Based on the AlphaFold modelling and the experimental results, our data support the pathogenicity of these mutations and substantiate the relevance of this interaction.

Our data are consistent with a core complex composed of ERLINs, RNF170, and TMUB1-L. Modelling predicts that both TMUB1-L and RNF170 bind the ERLIN scaffold at the same time; however, we cannot exclude the presence of different complexes containing one or the other protein. In contrast to RNF170, TMUB1 still binds RNF185 and TMEM259 in the absence of ERLINs. A previous study showed that the interaction of TMUB1 with RNF185 is not direct but mediated by TMEM259, and that ERLINs were not essential for ERAD of substrates regulated by RNF185 (van de Weijer et al, 2020), which thus is in agreement with our findings. In line with a recent publication (Wang et al, 2022), we confirmed the interaction of TMUB1-L with VCP, and we propose that it occurs via FAF2 (Figs 2G and S2B and C). FAF2 was also previously shown to interact with ERLIN2 upon overexpression (Wolf et al, 2021), and our data suggest that this interaction is mediated by TMUB1-L.

Oligomers of ERLINs form cup-shaped complexes that project into the lumen of the ER. It has been estimated that each ERLIN subunit can bind up to four molecules of cholesterol (Huber et al, 2013). Cells depleted of ERLINs show activation of SREBPs (Huber

et al, 2013), and we confirmed this finding. It was proposed that ERLINs can directly bind INSIG1 (Huber et al, 2013), a protein that retains SCAP in the ER upon cholesterol sufficiency, thereby suppressing the trafficking of SREBPs to the Golgi, where they are proteolytically processed to release an active transcription factor that up-regulates genes involved in cholesterol biosynthesis (Shimano & Sato, 2017). In this scenario, ERLINs would act as scaffolds where not only cholesterol is concentrated, but also where the INSIG1-SCAP interaction is stabilized. Our data provide an alternative explanation for how ERLIN scaffolds regulate cholesterol homeostasis. We propose that cholesterol bound to ERLINs is less accessible to SOAT1, preventing its esterification and facilitating cholesterol transport to the Golgi. Esterification of cholesterol and its accumulation in LDs together with triacylglycerols is a strategy by which cells detoxify excess cholesterol; however, in DKO cells, unhinged cholesterol esterification would lead to cholesterol depletion on the ER, triggering SCAP-mediated movement of SREBPs to the Golgi. In support of this model, we show that inhibition of SOAT1 with avasimibe rescues the accumulation of large LDs and blunts the up-regulation of the SREBP target gene HMGCS1 in DKO cells. Furthermore, avasimibe treatment restores the morphology of the Golgi apparatus in DKO, indicating that the Golgi vesiculation is most likely the result of cholesterol depletion in the Golgi.

Other data render our model plausible. First, cholesterol-binding sites have been mapped to residues located on the luminal side of the ERLIN cups (Hulce et al, 2013) so that cholesterol may remain buried within the cavity of the complex and inaccessible to SOAT1. Second, TMUB1 was also found to bind cholesterol (Hulce et al, 2013), and SOAT1 was present in the TMUB1 interactome, supporting the proximity of this enzyme to the complex. Notably, DKO cells showed a tendency to increased levels of SOAT1 ($log_2FC = 0.40$; q = 0.07), despite unchanged transcript levels. Third, the presence of ARF1 in the ERLIN2 IP supports the location of the ERLIN scaffolds to ER–Golgi contact sites involved in cholesterol trafficking. Taken together, our results illustrate the role of SPFH domain–containing proteins in defining specific membrane nanodomains where lipid metabolism activities are regulated, and to which proteins are recruited or excluded. Recently, an inhibitory relationship between calcium release from the ER via IP$_3$ receptors and cholesterol transport to the Golgi has been unravelled (Malek et al, 2021), rationalizing the two-pronged function of ERLINs to coordinate ERAD of IP$_3$ receptors via RNF170 (Lu et al, 2011) and cholesterol homeostasis.

Depletion of TMUB1-L isoform alone was sufficient to induce a Golgi fragmentation comparable to the depletion of the ERLIN complex. Intriguingly, TMUB1 and FAF2 are the closest orthologues to the yeast proteins Dsc3 and Ubx3, which are implicated in the so-called endosome and Golgi-associated degradation (Schmidt et al, 2019). This pathway regulates sphingolipid metabolism by actively extracting membrane proteins at the Golgi and delivering them to the proteasome. Whether a similar pathway exists in mammalian cells and involves TMUB1 is currently unknown. Several studies have linked Golgi vesiculation and trafficking of secretory proteins from the trans-Golgi network to cholesterol imbalance (Runz et al, 2006; Reverter et al, 2014). We used the model VSVG molecule to show that trafficking is halted at the Golgi, but the SILAC proteomics

suggest a redistribution of proteins that move along the secretory pathway, supporting a more global defect in DKO cells. This involves several adhesion molecules, explaining the migration and cell spreading defects of DKO cells, and implicating perturbations of these pathways in the degeneration of long corticospinal axons in HSP caused by mutations in *ERLIN1* or *ERLIN2*. Another form of HSP (SPG5) is caused by mutations in *CYP7B1* (Tsaousidou et al, 2008), which encodes a cytochrome P450 7α-hydroxylase implicated in cholesterol and bile acid metabolism. As a result, patients accumulate 25-hydroxycholesterol and 27-hydroxycholesterol (Schule et al, 2010). The latter are potent inhibitors of SREBPs and therefore can suppress cholesterol synthesis, potentially leading to cholesterol depletion in the Golgi (Radhakrishnan et al, 2007), and to a similar pathogenic cascade. In recent years, causative HSP mutations have been identified in *RNF170* (Wagner et al, 2019; Chouery et al, 2022). Future studies will be needed to fully understand the link between unbalanced cholesterol metabolism, regulated ubiquitination at the ER, and HSP.

# Materials and Methods

## Cell culture

HeLa cells were cultured at 37°C and 5% $CO_2$ in DMEM containing 4.5$g$/litre of glucose (# 11960044; Gibco), 2 mM glutamine (# 25030024; Gibco), 100 U/ml penicillin, 100 $\mu$g/ml streptomycin (# 15140122; Gibco), and 10% FetalClone III (#SH30109.03; GE Healthcare Hyclone). For SILAC labelling, cells were grown in arginine- and lysine-free DMEM with 4.5$g$/litre glucose (# 28001300; Silantes), containing dialysed FCS (#26400044; Gibco), penicillin–streptomycin–glutamine (# 10378016; Gibco), and different isotopes of arginine and lysine. We used medium isotopes (Arg6, Lys4) (# 201203902 and 211103913; Silantes) for the WT, light (Arg0, Lys0) (# 201003902 and 211003902; Silantes) for the DKO, and heavy (Arg10, Lys8) (# 201603902 and 211603902; Silantes) for the DKO$^{+E1/E2}$. Cells were regularly tested to be mycoplasma-free.

## Knock-out of *ERLIN1* and *ERLIN2* in HeLa cells via CRISPR/Cas9

*ERLIN1* and *ERLIN2* were knocked out using a CRISPR/Cas double-nickase approach as described previously (Ran et al, 2013). CRISPR guide RNAs (gRNAs) were designed to target exon 1 of *ERLIN1* (guide sequence sense strand 5′-GCCATCTGGCTGTGTACTAC-3′, guide sequence antisense strand 5′-TGTGGATGGAGGCGTAGAGC-3′) and exon 2 of *ERLIN2* (guide sequence sense strand 5′-GTTGGGAG-CAGTTGTGGCTG-3′, guide sequence antisense strand 5′-AGGA-TAAAGGCTCACTGATG-3′), respectively, and confirmed with CRISPOR (http://crispor.tefor.net). Oligonucleotides were purchased from Sigma-Aldrich and cloned into the pX335-U6-Chimeric_BB-CBh-hSpCas9n(D10A) vector, a gift from Feng Zhang (plasmid # 42335; Addgene) (Cong et al, 2013). Constructs were transfected into HeLa cells on five consecutive days, and clones were obtained via serial dilution. Positive clones were identified by Western blot analysis using anti-ERLIN1 and anti-ERLIN2 antibodies and confirmed by subcloning targeted gene regions into pcDNA3 (amplification of

exon 1 of *ERLIN1* with 5′-TTAGGATCCGCGAGAGAGCGCCAAGTTTC-3′ and 5′-AATGCGGCCGCCAACATCTGGAGGGTGCT-3′ and amplification of exon 2 of *ERLIN2* with 5′-TTAGGATCCGAAAGGGGAACGTTGACTGA-3′ and 5′-AATGCGGCCGCCTTTCCCGTGGTTACTGACA-3′) followed by sequencing.

## Generation of retroviruses and transduction

Coding regions of *ERLIN1* and *ERLIN2* (amplified with 5′-GGGGA-CAAGTTTGTACAAAAAAGCAGGCTAGAATGAATATGACTCAAGCCCG-3′ and 5′-GGGGACCACTTTGTACAAGAAAGCTGGGTCTCAACCTGTGCTCTCTTTGTTTT-3′; 5′-GGGGACAAGTTTGTACAAAAAAGCAGGCTACCATGGCTCAGTTGGGAGC-AGTT-3′ and 5′-GGGGACCACTTTGTACAAGAAAGCTGGGTCTCAATTCTCCTTA GTGGCCGTCT-3′, respectively) were cloned into the pDONR221 and inserted into the pBABE-puro vector using the gateway recombinant cloning (Morgenstern & Land, 1990). The pBABE-puro vector was a gift from Hartmut Land & Jay Morgenstern & Bob Weinberg (plasmid # 1764; http://n2t.net/addgene:1764; RRID:Addgene_1764; Addgene). Respective constructs were transfected into Phoenix-AMPHO packaging cells for 24 h. The harvested virus-containing medium was filtered through a 0.45-$\mu$m filter. For infection, HeLa cells at 70% confluency were incubated for 24 h in a virus-containing medium and 4 $\mu$g/ml polybrene. Cells with positive genome integration were selected by incubation of infected cells in DMEM containing 5 $\mu$g/ml puromycin for 48 h. The expression of the gene of interest was confirmed by Western blot analysis using ERLIN1 and ERLIN2 antibodies, respectively.

## TMUB1-L KO generation

The start codon of the *TMUB1* gene was mutagenized with an adenine base editor to convert the ATG into GTG (Gaudelli et al, 2017). The gRNA was designed using the rgenome webtool (http://www.rgenome.net/), and primers were purchased from Sigma-Aldrich and cloned into gRNA Cloning Vector, a gift from George Church (plasmid # 41824; Addgene) (Mali et al, 2013) after digestion with AflII and through the Gibson assembly of the PCR product of the primer pair (guide sequence sense strand 5′-TTTCTTGGCTTTATATATCTTGTGGA AAGGACGAAACACCGCGCCATGACCCTGATTGAAG-3′, guide sequence antisense strand 5′-GACTAGCCTTATTTTAACTTGCTATTTCTAGCTCTAAAACCTT CAATCAGGGTCATGGCGC-3′). The gRNA-containing vector was cotransfected with the pCMV_ABEmax_P2A_GFP, a gift from David Liu (plasmid # 112101; Addgene) (Koblan et al, 2018), and after 24 h sorted using INFLUX Cell-Sorter (BD Biosciences). Two 96-well plates were prepared, and the successful mutagenesis was tested via Western blot analysis.

## ERLIN2 overexpression constructs

pERLIN2-3xFLAG, pERLIN2-3xFLAG (R36K), pERLIN2-3xFLAG (G48V), pERLIN2-3xFLAG (H50Y), pERLIN2-3xFLAG (R36K-H50Y), and pERLIN2-3xFLAG (R36K-G48V-H50Y) were cloned in the p3XFLAG-CMV-14. All the mutations were confirmed via Sanger sequencing. For cloning, *ERLIN2* was amplified with 5′-TTAGAATTCATGGCTCAGTTGGGAGCAGTT-3′ and 5′-AATGGATCCATTCTCCTTAGTGGCCGTCTC-3′ from cDNA and the product was cloned into p3XFLAG-CMV-14 using EcoRI and BamHI. The R36K variant was generated with Q5 Site-Directed Mutagenesis Kit (#

E0554S; NEB) from the pERLIN2-3XFLAG using the manufacturer's protocol (primer sequence: Fwd 5'-GTATATTACAAAGGCGGTGCC-3' and Rev 5'-CCCAATATGTCCCTCTTCTATC-3'). All the other variants were generated using the Gibson assembly (HiFi DNA Assembly Master Mix, # E2621S; NEBuilder) from the pERLIN2-3XFLAG using the manufacturer's protocol. The pERLIN2-3XFLAG was amplified using four primers, two of which binding the backbone common for all the mutations (Fwd 5'-TAATGAGTGAGCTAACTCACATTAATTGCGTTGCG-3' and Rev 5'-GTGAGT-TAGCTCACTCATTAGGCACCCC-3'), and other two primers specific for the desired mutation (G48V: 5'-CAGCGGCCCTGTTTTCCATCTCATGCTCC-3' and 5'-GATGGAAAACAGGGCCGCTGGTCGAAGTC-3'; H50Y: 5'-CCCTGGTT-TCTATCTCATGCTCCCTTTCATC-3' and 5'-GCATGAGATAGAAACCAGGG-CCGCTGGTC-3'; G48V-H50Y: 5'-CAGCGGCCCTGTTTTCTATCTCATGCTCCCTTTC-3' and 5'-GATAGAAAACAGGGCCGCTGGTCGAAGTC-3').

### TMUB1 siRNA

For *TMUB1* down-regulation, stealth siRNA from Thermo Fisher Scientific was used at 100 nM final concentration. After 2 d from the first transfection, performed using Lipofectamine 2000, a second transfection was performed and the cells were analysed after 2 d. Shorter protocols failed to down-regulate TMUB1 in our system. Three siRNAs were used: 5'-ACCUCCCUCCCAACUGCGUUCUCCA-3', 5'-GGGAACAGCAGGUGCGACUCAUCUA-3', and 5'-GCCAUGGCAGCUACC-GACAGCAUGA-3'.

### Flow cytometry

Cell size was compared using BD LSRFortessa Analyzer. 24 h after seeding, cells were detached with trypsin, resuspended in fresh DMEM, and washed twice with PBS. Pellets were resuspended in PBS and analysed. Cell debris and cell doublets were excluded from the analysis by gating. For the TMUB1-L KO sorting, an untransfected control not expressing the GFP was used to discriminate the GFP-positive events.

### Scratch assay

$2 \times 10^6$ cells were seeded on a six-well plate and grown confluent over 3 d. A wound was introduced using a 1-ml pipette tip before the plate was washed with PBS and the fresh culture medium was added. Wound size was subsequently imaged with an Echo Revolve microscope. Plates were washed with PBS, and fresh culture medium was added, before imaging every 24 h. The area of wound size was quantified manually using Fiji/ImageJ 2.9.0 (Schindelin et al, 2012).

### Immunofluorescence staining

Cells were washed once with PBS 1X (# 18912014; Thermo Fisher Scientific) and fixed at RT for 15 min in 4% PFA, pH 7.4 (# P6148-500G; Sigma-Aldrich). After washing with PBS, cells were incubated for 10 min at RT in PBS containing 0.2% Triton X-100 (# 93418; Fluka) for permeabilization. Subsequently, cells were washed again and treated with 10% goat serum in PBS for 10 min at RT. Depending on the primary antibody, cells were incubated for 2 h with the antibody diluted in 1% goat serum in PBS. After three washing steps with PBS

for 5 min, cells were incubated for 1 h in a secondary antibody diluted in 1% goat serum in PBS. Cells were washed again three times for 5 min with PBS, whereas DAPI (# D9542-1MG; Sigma-Aldrich) was added at a final concentration of 1.25 $\mu$M during the second wash. Samples were mounted with FluorSave reagent (# 345789; Millipore). The antibodies used in this study are the following: anti-ERLIN2 (1:1,000, 2959; Cell Signaling), anti-TMUB1 (1:1,000, ab180586; Abcam), anti-GM130 (1:1,000, 610822; BD Biosciences), Alexa Fluor 488 anti-rabbit IgG (1:1,000, # A-11008; Thermo Fisher Scientific), and Alexa Fluor 594 anti-mouse IgG (1:1,000, # A-11012; Thermo Fisher Scientific). For LD staining, no permeabilization was performed and Bodipy 493/503 (D3922; Thermo Fisher Scientific) was applied just after fixation with PFA 4% for 30 min at a final concentration of 5 $\mu$M and washed five times for 10 min in PBS.

### Transfection and live-cell imaging

Transfection of cells was performed using Lipofectamine 2000 (# 11668019; Invitrogen), 24 h after seeding. Before the transfection, the medium of the cells was changed to OptiMEM. For the transfection, 2 $\mu$l of Lipofectamine was diluted in 250 $\mu$l of OptiMEM (# 11058021; Gibco) and kept for 5 min at RT. Afterward, the Lipofectamine: OptiMEM mix was added to 250 $\mu$l of OptiMEM containing 1 $\mu$g of DNA. After 15 min, the DNA:Lipofectamine complex was added dropwise to the cells, and after 6-h incubation, the medium was exchanged by DMEM. Live imaging was performed 24 h after the transfection and at 37°C with 5% $CO_2$ to resemble the incubator conditions. The RUSH system was used to analyse ER-to-Golgi trafficking similarly as described before (Boncompain et al, 2012). Cells were transfected 24 h before imaging with Str-Ii_VSVG-SBP-EGFP, a gift from Franck Perez (plasmid # 65300; http://n2t.net/addgene:65300; RRID:Addgene_65300; Addgene). Live imaging was performed with a spinning disc confocal microscope (Ultra*VIEW* VoX; PerkinElmer) for 45 min, and one frame was acquired every 30 s using Volocity software. Imaging was performed using a 60x oil objective (NA 1.49), and acquisition of the video was initiated directly after the biotin was added to the medium.

### Image analysis

Image analysis was performed using Fiji/ImageJ 2.9.0 (Schindelin et al, 2012). ER morphology was quantified as previously described (Hu et al, 2009). By visual inspection, cells were classified into three categories (normal, intermediate, or collapsed) according to the ER shape using RTN4 immunostaining. The RTN4 area was calculated using the same staining after a Gaussian blur (Sigma = 20) filter was applied. The area was later obtained by thresholding with fixed values for each biological replicate. For the quantification of the Golgi apparatus and LDs, after manual segmentation of the cells, the Golgi apparatus and LDs were automatically segmented using GM130 immunostaining and Bodipy staining, respectively. Segmentation was performed using a fixed threshold for all genotypes in each biological replicate. The objects derived by thresholding were counted per cell using the "Analyze particles" function available in Fiji. For VSVG-GFP/GM130 colocalization, after subtraction of the background calculated from an area outside the

cells, the cells were manually segmented, and for each cell, both the total VSVG-GFP fluorescence intensity and the fluorescence intensity within a mask obtained from GM130 signal thresholding were calculated. The value used is the ratio between the fluorescence intensity inside the Golgi and the total fluorescence intensity for each cell. The scale bar is equal to 10 $\mu m$.

### RNA isolation, RNA-seq, and qRT-PCR analysis

RNA of cells was isolated with TRIzol reagent (# 15596026; Invitrogen), according to the manufacturer's protocol (Life Technologies). Reverse transcription was performed using SuperScript First-Strand Synthesis System (# 11904018; Thermo Fisher Scientific). Real-time qRT-PCR was performed in a thermocycler (Applied Biosystems) with SYBR Green Master Mix (# 4309155; Thermo Fisher Scientific) and respective primer pairs (*ERLIN1* sense primer 5′-GAAAGCTCACTCCCCTC-TAAG-3′ and antisense primer 5′-TGTTCCCACTTAACCCCTTG-3′; *ERLIN2* sense primer 5′-ACGCTTCAAGAGGTCTACATTG-3′ and antisense primer 5′-ATTGCCTCTGGTATGTTGGG-3′; *HMGCS* sense primer 5′-CATTA-GACCGCTGCTATTCT-3′ and antisense primer 5′-AGCCAAAATCATT-CAAGGTA-3′). *GAPDH* was used as a reference gene in all analyses (GAPDH sense primer 5′-AATCCCATCACCATCTTCCA-3′ and antisense primer 5′-TGGACTCCACGACGTACTCA-3′), and the fold enrichment was calculated using the formula 2(−ΔΔCt). For RNA-seq analysis, the RNA was submitted to the Cologne Center for Genomics. Libraries were prepared using the Illumina stranded TruSeq RNA sample preparation kit. ERCC RNA Spike-In Mix (Thermo Fisher Scientific) was added to the samples before library preparation. Library preparation started with 1 $\mu g$ total RNA. After poly-A selection (using poly-T oligo-attached magnetic beads), mRNA was purified and fragmented using divalent cations under elevated temperature. The RNA fragments underwent reverse transcription using random primers. This was followed by second-strand cDNA synthesis with DNA polymerase I and RNase H. After end repair and A-tailing, indexing adapters were ligated. The products were then purified and amplified (15 PCR cycles) to create the final cDNA libraries. After library validation and quantification (Agilent Tape-Station), equimolar amounts of the library were pooled. The pool was quantified using the Peqlab KAPA Library Quantification Kit and Applied Biosystems 7900HT Sequence Detection System. The pool was sequenced on an Illumina NovaSeq 6000 sequencing instrument with a PE100 protocol.

### Analysis of RNA-seq

RNA-seq was performed with a directional protocol. Quality control, trimming, and alignment were performed using the nf-core (Ewels et al, 2020) RNA-seq pipeline (v3.0) (10.5281/ZENODO.1400710). Details of the software and dependencies for this pipeline can be found at the referenced DOI for the pipeline and at https://github.com/nf-core/rnaseq/blob/master/CITATIONS.md. The reference genome sequence and transcript annotation used were *Homo sapiens* genome GRCh38 from Ensembl version 103. Differential expression analysis was performed in R version 4.1.2 (2021-11-01) (R core Team, 2021) with DESeq2 v1.34.0 (Love et al, 2014) to make pairwise comparisons between groups. Log fold change shrinkage estimation was performed with ashr (Stephens, 2016). Only genes

with a minimum coverage of 10 reads in all samples from each pairwise comparison were considered as candidates to be differentially expressed. Genes were considered differentially expressed if they showed a $|log_2(fold\ change)| > 1$ and were below an FDR of 0.05. Genes with a minimum coverage of 10 reads in all samples from each pairwise comparison were included in functional enrichment analyses and considered as the "gene universe" for over-representation–based analyses. Functional enrichment analysis was performed with clusterProfiler v4.2.0 (Yu et al, 2012).

### Protein extraction and immunoblotting

Cells were harvested by scraping, and the pellet was lysed in an equal volume of RIPA buffer (50 mM Tris–HCl, pH 7.4, 150 mM NaCl, 1 mM EDTA, 1% Triton X-100, 0.25% sodium deoxycholate, 1 mM sodium orthovanadate) containing 1X Protease Inhibitor Cocktail (# P2714-1BTL; Sigma-Aldrich) for 30 min on ice. Samples were centrifuged for 30 min at 20,000$g$ at 4°C, and protein concentrations of the supernatants were measured with a standard Bradford assay (Bio-Rad Laboratories). 25 or 50 $\mu g$ of proteins was diluted in Laemmli buffer 3X (188 mM Tris–HCl, 6% SDS, 12% glycerol, 0.05% bromophenol blue, 10% $\beta$-mercaptoethanol) and then analysed by SDS–PAGE followed by immunoblot analysis. Proteins were blotted on a polyvinylidene difluoride (PVDF) membrane (#10600023; Amersham) of pore size 0.45 $\mu m$ using wet chambers and blotting buffer (25 mM Tris, 192 mM glycine, 20% methanol) at 4°C for either 1.5 h at 300 mA or for 16 h at 30 V. After blotting, proteins were stained for 10 min in Ponceau S. Membranes were blocked for 30 min in 5% skim milk (T145.3; Carl Roth) in TBST (20 mM Tris–HCl, pH 7.4, 150 mM NaCl, 0.1% Tween-20) at RT. Primary antibodies were diluted in the same blocking solution and incubated overnight at 4°C or for 2 h at RT. Respective HRP-linked secondary antibodies were used at 1:10,000 dilution and incubated for 1 h at RT. After incubation with both primary and secondary antibodies, membranes were washed three times in TBST. HRP was detected using ECL Western blotting detection reagents (# RPN2106; VWR). Chemiluminescence was detected using X-ray films (# 4005194; Bema) or the Vilber Fusion Solo S. Intensities of bands of interest were quantified using ImageJ. The antibodies used in this study are the following: anti-ERLIN1 (1:1,000, HPA011252; Sigma-Aldrich), anti-ERLIN2 (1:1,000, 2959; Cell Signaling), anti-TMUB1 (1:1,000, ab180586; Abcam), anti-TMUB2 (1:1,000, 28044-1-AP; Proteintech), anti-GAPDH (1:2,000, #MAB374; Millipore), anti-flotillin-1 (1:1,000, #610821; BD Transduction Laboratories), anti-KIDINS220 (1:1,000, PA5-22116; Thermo Fisher Scientific), anti-RTN4 (1:1,000, NB100-56681; Novus Biologicals), anti-VDAC2 (1:1,000, 9412S; Cell Signaling), anti-TMEM259 (1:1,000, HPA042669; Atlas Antibodies), anti-PMP70 (1:1,000, ab85550; Abcam), anti-FLAG (1:5,000, # F7425; Sigma-Aldrich), HRP-linked anti-mouse IgG (1:10,000, #A9044; Sigma-Aldrich), HRP-linked anti-rabbit IgG (1:10,000, #A0545; Sigma-Aldrich).

### Immunoprecipitation

1.5 × 10$^7$ cells were seeded on a 15-cm dish 1 d before the lysis. The next day, cells were washed once with PBS at RT, harvested with a cell scraper, pelleted at 60$g$ for 3 min, and resuspended in 400 $\mu l$ IP buffer (Tris–HCl, pH 7.4, 50 mM, KCl 50 mM, Triton X-100 0.5%)

containing 1X Protease Inhibitor Cocktail (# P2714-1BTL; Sigma-Aldrich). Then, cells were passed six times through a 29 Gauge needle, incubated for 30 min on ice, and centrifuged at 20,000$g$ for 30 min at 4°C. Protein concentration was determined via the Bradford assay, 500 $\mu$g of proteins was transferred in a new microcentrifuge tube, and the volume was adjusted to 250 $\mu$l using IP buffer. Next, 0.5 $\mu$g of the respective antibody was added to the proteins and incubated overnight at 4°C on a rotor. An anti-RFP antibody (600-401-379S; Rockland) was used as a control antibody for the TMUB1 IP. Afterward, 20 $\mu$l of Dynabeads Protein G (# 10003D; Thermo Fisher Scientific) was equilibrated three times with 1 ml of ice-cold IP buffer using a magnetic rack, added to the protein:antibody lysate, and incubated for 3 h at 4°C. After incubation, the magnetic rack was used to separate the beads from the flow-through and beads were washed three times with 1 ml of IP buffer. Depending on the followed application, elution was performed either in 30 $\mu$l of Laemmli buffer 3X, for Western blot analysis, or in SP3 buffer (5% SDS in 1X PBS) for proteomics analysis. After the addition of the respective elution buffer, beads were vortexed for 1 min and incubated at 95°C for 5 min. The eluate was separated from the beads using the magnetic rack.

### DRM isolation

DRMs were isolated according to the protocol from George and colleagues (George et al, 2010) with the following changes. Lipid–protein crosslinking step was not performed, and the Triton X-100 concentration of the raft isolation buffer contained 5% Triton X-100, 150 mM NaCl, 5 mM EDTA, 50 mM Tris–HCl, pH 7.4. Per sample, 1.5 × 10$^7$ cells were seeded on 15-cm plates, after 2 d washed twice with cold PBS + 5 mM EDTA, harvested using a cell scraper, and centrifuged for 10 min at 11,200$g$ and at 4°C. Cell pellets were kept overnight at –80°C. Samples were washed once with TBS and lysed in raft isolation buffer. After lysis, cell lysates were centrifuged for 15 min at 112$g$, 4°C, and protein concentrations of the supernatants were determined via the Bradford assay. 1 mg of proteins was diluted in 500 $\mu$l raft isolation buffer and subsequently mixed with 1 ml 60% OptiPrep (# 1114542; Axis-Shield) solution to obtain a final concentration of 40% iodixanol. Homogenates were transferred into ultra-clear tubes (# 344062; Beckman and Coulter) and overlaid with a step gradient of 30% iodixanol solution and 5% iodixanol solution to a final volume of 4 ml. The step gradients were centrifuged for 5 h at 132,000$g$ at 4°C (SW60# 335649; Beckman and Coulter). Subsequently, five fractions were obtained by taking 800 $\mu$l from the top to bottom.

### Proteomics analysis of IPs, post-nuclear lysate, and SILAC-labelled DRMs

#### Sample preparation
After IP, the proteins eluted in SP3 buffer were reduced, alkylated, and digested with trypsin using the SP3 protocol originally described by Hughes et al (2019). For the analysis of DRM fractions generated from SILAC-labelled cells and label-free profiling of the post-nuclear lysate, the proteins were precipitated with four times the volume of ice-cold acetone by incubation at –80°C for 15 min followed by 2 h at –20°C. After centrifugation at 16,000$g$ for 15 min at

4°C, the supernatant was discarded, and the pellet was washed with $\mu$l of ice-cold acetone and centrifuged twice for 5 min at 16,000$g$. The air-dried pellet was resuspended in 8 M urea in 50 mM triethylammonium bicarbonate (TEAB), pH 8.5, and digested with LysC and trypsin as described previously (Schatton et al, 2022). Digested samples were loaded on Stage Tips and submitted to the CECAD proteomics facility.

#### Data acquisition
Immunoprecipitated and SILAC-labelled samples were analysed in a data-dependent acquisition mode (DDA) on a Q Exactive Plus mass spectrometer (Thermo Fisher Scientific) as described previously (Schatton et al, 2022). For whole proteome profiling of the post-nuclear lysate, samples were analysed in a data-independent acquisition mode (DIA) on a Q Exactive Plus Orbitrap mass spectrometer (Thermo Fisher Scientific) that was coupled to an EASY-nano-LC 1000 system (Thermo Fisher Scientific). Samples were loaded onto an in-house–packed analytical column (50 cm–75 $\mu$m I.D., filled with 2.7 $\mu$m Poroshell 120 EC-C18, Agilent) equilibrated in buffer A (0.1% formic acid in water). Peptides were chromatographically separated with a flow rate of 250 nl/min and a 110-min gradient, followed by a 10-min column wash with 95% solvent B (0.1% formic acid in 80% acetonitrile). To generate an experiment-specific gas phase–fractionated library, a pool of all samples was used for narrow window DIA measurements covering the range from 400 to 1,000 m/z (Searle et al, 2020) with six consecutive 100 m/z acquisitions. MS1 scans of the 100 m/z gas phase fraction were acquired at 30-k resolution. The maximum injection time was set to 50 msec and the AGC target to 3 × 10$^6$. MS2 scans were acquired in 25 × 4 m/z staggered windows resulting in 50 nominal 2 m/z windows after demultiplexing. MS2 settings were 17,500 resolution, 60 msec maximum injection time, and an AGC target of 1 × 10$^6$. For the acquisition of the sample data, an identical chromatography method was used. MS1 scans were acquired at 35,000 resolution, the maximum injection time was set to 60 msec, and the AGC target was set to 1 × 10$^6$. DIA scans covering the precursor range from 400 to 1,000 m/z and were acquired in 25 × 25 m/z staggered windows. MS2 spectra starting at 200 m/z were acquired at 17,500 resolution with a maximum injection of 60 msec and an AGC target of 1 × 10$^6$.

#### Data processing
DDA raw data were processed with MaxQuant version 1.5.3.8 or 2.0.1.0 (Cox & Mann, 2008; Cox et al, 2011) as described previously (Schatton et al, 2022). LFQ quantification was enabled with default settings, and the match-between-runs option was enabled within replicate groups. For SILAC ratio quantification, the label multiplicity was set to 3 with Arg6 and Lys4 selected as medium and Arg10 and Lys8 as heavy labels. DIA data were analysed with DIA-NN 1.8.1 (Demichev et al, 2020). First, the narrow window DIA runs were demultiplexed and transformed to mzML files using the msconvert module in ProteoWizard. These files were used to generate an experiment-specific spectral library from a SwissProt human canonical fasta file using the library generation option in DIA-NN. The settings differing from default were as follows: Min precursor m/z set to 400, Max precursor m/z set to 1,000, heuristic protein inference, and no shared spectra. The experiment-specific spectral

library (10,471 protein isoforms, 11,109 protein groups, and 104,704 precursors) was used for subsequent sample analysis. For statistical analysis, the MaxQuant or DIA-NN output tables were imported to Perseus version 1.6.15.0 (Tyanova et al, 2016). Potential contaminants were removed, intensities were $\log_2$-transformed, and the datasets were filtered for at least 100% data completeness in at least one condition. The remaining missing values were imputed with random values from the lower end of the intensity distribution using Perseus defaults. t tests and ANOVAs were calculated with permutation-based FDR control (q < 0.05). For the five DRM fractions, no imputation was used and the following steps were performed separately. Significant candidates with changing SILAC ratios between any conditions were identified by ANOVA multiple sample testing (S0 = 0.1, permutation-based FDR < 0.05 with 250 randomizations). Next, the dataset was filtered for significantly regulated candidates, SILAC ratios were Z-score normalized protein-wise, and Euclidean hierarchical clustering was performed to identify protein clusters with similar changing protein SILAC ratios over the three conditions. Finally, all fractions were merged, protein groups were annotated with gene ontology (GO) terms and compartments (Itzhak et al, 2016), and Fisher's exact tests were performed for each fraction separately (Benjamini–Hochberg FDR = 0.02) to identify regulated processes systematically.

## Lipidomics

### Triacylglycerols and cholesteryl esters
Cells were homogenized in Milli-Q water (~$10^6$ cells/100 μl) using the Precellys 24 Homogenizer (Peqlab) at 6,500 rpm for 30 s. The protein content of the homogenate was routinely determined using bicinchoninic acid. To 50 μl of cell homogenate, 450 μl of Milli-Q water, 1.875 ml of chloroform/methanol/37% hydrochloric acid 5:10:0.15 (vol/vol), and internal standards (30 μl of d5-TG Internal Standard Mixture I and 256 pmol cholesteryl ester 19:0; Avanti Polar Lipids) were added. Lipid extraction was performed as previously described (Kumar et al, 2015). Triacylglycerols and cholesteryl esters were analysed by nano-electrospray ionization tandem MS (nano-ESI-MS/MS) with direct infusion of the lipid extract (Shotgun Lipidomics) as previously described (Özbalci et al, 2013; Hammerschmidt et al, 2019). Endogenous lipid species were quantified by referring their peak areas to those of the respective internal standard. The calculated lipid amounts were normalized to the protein content of the cell homogenate.

### Cholesterol
Cholesterol levels in cells were determined by liquid chromatography coupled to electrospray ionization tandem mass spectrometry (LC-ESI-MS/MS): to 30 μl of the cell homogenate mentioned above, 70 μl of Milli-Q water and 1.26 nmol d7-cholesterol as internal standard (Avanti Polar Lipids) were added. Lipid extraction and LC-MS/MS analysis of cholesterol were performed as previously described (Mourier et al, 2015).

### Ceramides and sphingomyelins
Levels of ceramides and sphingomyelins in isolated DRMs were determined by LC-M/MS by treating the samples using a procedure previously described (Ejsing et al, 2009) with some modifications:

to 70 μl of DRM preparation, 130 μl of 155 mM ammonium carbonate buffer was added. Lipids were extracted by adding 990 μl of chloroform/methanol 17:1 (vol/vol) and internal standards (127 pmol ceramide 12:0 and 124 pmol sphingomyelin 12:0; Avanti Polar Lipids), followed by shaking at 900 rpm/min in a ThermoMixer (Eppendorf) at 20°C for 30 min. After centrifugation (12,000g, 5 min, 4°C), the lower (organic) phase was transferred to a new tube, and the upper phase was extracted again with 990 ml chloroform/methanol 2:1 (vol/vol). The combined organic phases were dried under a stream of nitrogen. The residues were resolved in 100 μl of Milli-Q water and 750 μl of chloroform/methanol 1:2 (vol/vol). Alkaline hydrolysis of glycerolipids and LC-ESI-MS/MS analysis of ceramides and sphingomyelins were done as previously published (Oteng et al, 2017).

## AlphaFold-Multimer prediction of the ERLIN1, ERLIN2, TMUB1, and RNF170 complex

Multimer predictions of the ERLIN1-ERLIN2-TMUB1-RNF170 complex were performed using Cologne High Efficient Operating Platform for Science (CHEOPS) provided by the Regional Computing Center of Cologne (RRZK) using 4 CPU cores of a node with 2x Intel(R) Xeon(R) Gold 6248 CPU @ 2.50 GHz, 60 GB RAM, and Nvidia Tesla V100-SXM2 GPU with 32 GB VRAM running the AlphaFold module 2.2.0 (Evans et al, 2021 Preprint; Jumper et al, 2021), with 5 seeds per model generating in total 25 models. Peptide sequences of ERLIN1 (sp|O75477|ERLN1_HUMAN Erlin-1), ERLIN2 (sp|O94905|ERLN2_HUMAN Erlin-2), TMUB1 (sp|Q9BVT8|TMUB1_HUMAN transmembrane and ubiquitin-like domain–containing protein 1), RNF170 (sp|Q96K19|RN170_HUMAN E3 ubiquitin–protein ligase RNF170), VCP (sp| P55072|TERA_HUMAN Transitional endoplasmic reticulum ATPase), and FAF2 (sp| Q96CS3|FAF2_HUMAN FAS-associated factor 2) were extracted from Uniprot.org. The top-ranked model, with the highest confidence, of the Multimer prediction was used for further analysis. AlphaFold models were visualized using the latest version of ChimeraX 1.5, and hydrogen bonds were predicted using the ChimeraX function "H-Bonds."

## Statistical analysis

A one-way ANOVA followed by post hoc Tukey's test was used to compare multiple groups. The statistical analysis of the MS analysis is described in detail in the "Data processing" section of the proteomics section. The number of biological replicates, tests used, and P-values are specified in the figure legends.

# Data Availability

All data supporting the findings of this study are available within the article and its Supplementary Information. Proteomics raw data are available via ProteomeXchange with identifier PXD048503.

# Supplementary Information

# Acknowledgements

The authors thank Esther Barth, the CECAD proteomics, lipidomics, imaging, and FACS facilities for support, and members of the Rugarli laboratory for constructive discussions. This work was funded by the Deutsche Forschungsgemeinschaft (RU 1653/4-1) and by the Walter and Monika Neupert Foundation to M Veronese. EI Rugarli and M Veronese are members of the RTG-Reloc (411422114-GRK 2550) funded by the DFG. The authors declare no competing financial interests.

## Author Contributions

M Veronese: conceptualization, formal analysis, investigation, visualization, and writing—original draft, review, and editing.
S Kallabis: formal analysis, visualization, and writing—review and editing.
A Tobias Kaczmarek: formal analysis, investigation, visualization, and writing—review and editing.
A Das: investigation, visualization, and writing—review and editing.
L Robers: investigation, visualization, and writing—review and editing.
S Schumacher: investigation, visualization, and writing—review and editing.
A Lofrano: investigation.
S Brodesser: formal analysis, investigation, and writing—review and editing.
S Müller: formal analysis and writing—original draft, review, and editing.
K Hofmann: conceptualization, investigation, and writing—review and editing.
M Krüger: conceptualization, supervision, and writing—review and editing.
EI Rugarli: conceptualization, supervision, funding acquisition, and writing—original draft, review, and editing.

## Conflict of Interest Statement

The authors declare that they have no conflict of interest.

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
