## [Reviewer comments · Life Science Alliance]

Life Science Alliance

ERLINs bridge TMUB1 and RNF170, limit cholesterol esterification, and regulate the secretory pathway

Matteo Veronese, Sebastian Kallabis, Alexander Kaczmarek, Anushka Das, Lennart Robers, Simon Schumacher, Alessia Lofrano, Susanne Brodesser, Stefan Müller, Kay Hofmann, Marcus Krüger, and Elena Rugarli

DOI: <https://doi.org/10.26508/lsa.202402620>

Corresponding author(s): *Elena Rugarli, University of Cologne*

Review Timeline:

Submission Date:	2024-01-25
Editorial Decision:	2024-02-26
Revision Received:	2024-04-30
Editorial Decision:	2024-05-03
Revision Received:	2024-05-06
Accepted:	2024-05-07

Scientific Editor: *Eric Sawey, PhD*

Transaction Report:

February 26, 2024

Re: Life Science Alliance manuscript #LSA-2024-02620-T

Prof. Elena I Rugarli
CECAD Research Center
Institute for Genetics
Joseph-Stelzmann-Str. 26
Koeln 50931
Germany

Dear Dr. Rugarli,

Thank you for submitting your manuscript entitled "ERLIN1/2 scaffolds bridge TMUB1 and RNF170 and restrict cholesterol esterification to regulate the secretory pathway" to Life Science Alliance. The manuscript was assessed by expert reviewers, whose comments are appended to this letter. We invite you to submit a revised manuscript addressing the Reviewer comments.

Thank you for this interesting contribution to Life Science Alliance. We are looking forward to receiving your revised manuscript.

Sincerely,

B. MANUSCRIPT ORGANIZATION AND FORMATTING:

Reviewer #1 (Comments to the Authors (Required)):

Dear Authors of the manuscript,

The manuscript is largely composed of two main parts. The first part explores how Erlins function as nanodomains, interacting with various proteins involved in protein degradation, such as the E3 ligase RNF170, structural components like TMUB1 and FAF2, and cholesterol. Erlins DKO provides a role in retention of TMUB1 in nanodomain of DRM. This interaction holds significance in the context of human diseases associated with Erlin variants, as the mutant form of Erlin2 fails to associate with TMUB1. The second part of the manuscript delves into the consequences of Erlin DKO. This disruption leads to cholesteryl ester (CE) accumulation, though not in free cholesterol. Additionally, there is an increase in triacylglycerol (TAG), lipid droplet (LD) levels rise, ultimately resulting in tubular ER collapses and Golgi fragmentation. These findings are supported by a series of proteomics and RNA seq analyses.

Interestingly, the authors noted that the activation of the SREBP pathway upon ERLINs depletion was milder than previously reported by Huber et al. This discrepancy could be attributed to differences in the cell types used in this study or, alternatively, the collapse of ER tubules and Golgi fragmentation induced by Erlins DKO might trigger accumulation of HMG-CoA reductase (Jo et al) by slower degradation of the protein, which leads to the increases of cholesterol and CE by SOAT activity. Also, Huber et al demonstrated the upregulation of nuclear SREBP, which resulted in the increases of free fatty acids and TAG synthesis. These observations are well reproduced in this manuscript.

The observation of ER and Golgi structural abnormalities is intriguing if these phenotypes are observed in clinical mutation conditions. These structural abnormalities were restored by the SOAT1 inhibitor, Avasimibe. This suggests that the increases in cholesteryl esters (CE) and lipid droplets (LD) caused abnormal ER and Golgi phenotypes in the absence of Erlins and stimulated SOAT1 activity. The underlying mechanism points to aberrant cholesterol synthesis, typically converted to cholesterol esters by SOAT1, especially under increased free cholesterol biosynthesis probably in part resulting from HMGR and/or nuclear SREBP stabilization following Erlins knockdown or knockout. There is another possibility that the increased CE but not cholesterol derived from the uptake of lipoprotein receptor mediated endocytosis since the culture condition was no depletion of lipoprotein used.

It provides valuable insights into the role of nanodomains composed of Erlins and other interacting proteins and molecules. The manuscript demonstrates their impact on the regulation of cholesterol and triglyceride metabolism, as well as their role in maintaining ER and Golgi structure. Additionally, it suggests a potential link to cell migration, highlighting the broader implications of these findings.

It raises some questions.

- Have the authors had a chance to check the protein levels of HMG-CoA reductase and nuclear SREBPs?
- Fig6A shows collapsed ER tubules by RTN4 staining. But the intensity of the green fluorescence seems much greater in the DKO. If the protein RNT4 levels provided together with the image would be better to understand the hypothesis.
- Fig8B shows GM130 in DKO low magnification, providing half of the cells in the field have quite a normal range of Golgi network except the magnified cell. Also, GM130 biochemical protein data inclusion may be helpful if the total Golgi content has been changed or not.
- Page 10, "~ increased cholesterol esterification maintains low levels of cholesterol in the ER" the author already stated that there was no change in unesterified cholesterol in the previous paragraph. There is a conflict of statements.
- it would be great if any secretory vesicle proteins (biochemical data) are restricted to the fragmented Golgi compared to the normal ER or Golgi network. One possibility would be reduction of GRAP55 /65 protein since it is known with the fragmented Golgi (<https://doi.org/10.1186/s13024-022-00568-2>). Also, this phenomenon may be related to the human diseases HSP, ALS or other protein depot accumulation phenomenon. It would be better to discuss the possible relationship between ER tubule collapses, Golgi morphology changes in Erlins DKO associated with diseases if anything known.
- Also, it may leave a question if those fragmented Golgi is associated with lysosomes/peroxisome. Have you checked the lysosome/peroxisome structure as a control or colocalization?
- The mutant of G48V in Erlin2 is very interesting since any mutation construct containing G48V shows disappearance of the bands, probably destabilization of the protein. However, it is curious why it was made in Erlin2 although the clinical mutation of G to V occurs in Erlin1 (Navarino, G et al. 2014). It is mentioned that G50V in Erlin1 is related to the SPG62. It is questionable if the authors have observed the collapsed ER tubules or Golgi fragmentation in these cells. It may be directly related to human

diseases.

Minor changes,

- It seems better to follow and understand if the figure 2i, and 2j goes to Fig 3 at first since the paragraph in the results goes with these together.
- reference page 26, Jo et al are same papers. Needs to be corrected.
- Material Methods, correction period to comma for page 19
 - o 11.200 x g -> 11,200 x g
 - 112 x g : is this really 112 x g or typo?
 - 132.000 x g -> 132,000 x g
- What are these abbreviations stand for GOMF , FFAT, VAPA ? They -latter two of them- look like directly from the reference.
- Page 9, the past paragraph needs a reference for "Esterification of excess cholesterol at the ER and its deposition in LDs are observed when cholesterol transport from the ER to the Golgi is impaired". Is this using ER to Golgi transport inhibited by a specific inhibitor?

Reviewer #2 (Comments to the Authors (Required)):

The well written an organized manuscript reports an ambitious and extensive molecular cell biology study to assess the cellular functions of the endoplasmic reticulum ERLIN1 and -2 proteins. The work is of high quality and the data clearly and professionally displayed. The study employs advanced up-to-date methodology, including e.g. CRISPR-Cas9 mediated genetic manipulations of the genes of interest, modeling of proteins and protein complexes with AlphaFold, and proteomics as well as lipidomics. The main findings suggest that ERLINs mediate interactions of the long isoform of TMUB1 and RNF170, and identify a luminal N-terminal conserved motif in both proteins required for this interaction. The authors further show quite convincing data that the ERLIN scaffolds play a role in maintaining ER cholesterol levels by favouring cholesterol transport to the Golgi over esterification, thereby regulating Golgi morphology and secretory pathway function. In my opinion the data presented are strongly supportive for each main point of the paper.

I have only minor comments:

1. p. 10, top paragraph, referring to Fig. 8E. The authors state that avasimibe treatment reverted the effect of ERLIN DKO on the HMGCS mRNA. However, when looking at panel 8E, the avasimibe-induced elevation is still there, even though the expression levels are markedly lower than in the DMSO control. Please comment and rewrite.
2. p. 11, end of 2nd paragraph: '...however our data strongly argue against a direct interaction.' Please specify which data strongly argue against this.
3. p. 12, middle paragraph: 'Third, the presence of Arf1 and Arf4...'. This statement is relatively weak since the Arfs have not been primarily described as membrane contact site components. They were initially identified as Golgi vesicle transport machinery and later on in many other functions such as mRNA transport, mTORC1 activity and mitochondrial dynamics. The only Arf function associated with the present topic is that of Arf1 OSBP binding on trans-Golgi membranes. Please rewrite.

Typos:

1. p. 5, line 2 from the bottom: '...and used it AS a control...'
2. p. 11, line 11: '...of unknowN significance in...'

Point-to-point response to the Reviewers' comments

Reviewer # 1

Major comments:

- *Have the authors had a chance to check the protein levels of HMG-CoA reductase and nuclear SREBPs?*

The protein levels of HMGCR have indeed been measured by proteomics and found to be significantly increased in DKO compared to WT (see Figure 4C), as also mentioned in the text. The exact measurements in four biological replicates and relative statistics can be found in the supplementary Table 4. We believe that these mass spectrometry measurements are more reliable in quantifying differences than a western blot.

Transcriptomic analysis (supplementary Table 3, and figure 4A) demonstrated a mild but significant increase of SREBPs-dependent transcripts, and this was confirmed by proteomics (Figure 4C). We have now performed western blot analysis using anti-SREBP1 antibody (see below). We do not detect a consistent increase in the amount of the nuclear form of the transcription factor (see Figure 1 for Reviewers). However, we believe that this is a limitation of the technique when we aim to measure small differences. Therefore, we prefer not to add these data to the manuscript. As the Reviewer pointed out, the difference with the previous study may depend on the fact that we are studying a KO in comparison to an acute downregulation of ERLINs.

[Figure removed by editorial staff per authors' request].

- *Fig6A shows collapsed ER tubules by RTN4 staining. But the intensity of the green fluorescence seems much greater in the DKO. If the protein RNT4 levels provided together with the image would be better to understand the hypothesis.*

The levels of RTN4 do not change between WT and DKO, as shown by proteomics analysis included in the manuscript. The relative data with statistics can be found in supplementary Table 4.

We now mention this explicitly in the text (page 8, first paragraph): “While the total levels of RTN4 did not change in cells of different genotypes (supplementary Table 4), immunofluorescence analysis showed altered ER morphology in DKO cells, which were intriguingly depleted of ER peripheral tubules in comparison to both WT and DKO^{+E1/E2} cells (Fig. 6A, B). “

The impression that RTN4 fluorescence may be increased in DKO comes from the different morphology of the ER in DKO cells, with accumulation of sheets at the cell periphery. We have also performed a western blot using RTN4 antibodies that confirm the proteomics data. We attach it below for the Reviewers.

[Figure removed by editorial staff per authors' request].

• Fig8B shows GM130 in DKO low magnification, providing half of the cells in the field have quite a normal range of Golgi network except the magnified cell. Also, GM130 biochemical protein data inclusion may be helpful if the total Golgi content has been changed or not.

More cells than the one shown in the enlargement have a fragmented Golgi, as demonstrated by the quantification, which was performed blind to the genotype and treatment using an automated macro. Similarly to what is stated above, also GM130 (GOLGA2) was measured in the proteomics (supplementary Table 4) and found unaffected. However, we find that some Golgi proteins, including GM130 and GRASP55 changed their distribution in the different cellular fractions analysed by SILAC (supplementary Table 5), further supporting Golgi fragmentation.

We have now added a sentence to comment on these data (page 8, last paragraph): “We then examined if Golgi proteins changed their abundance in the different fractions analysed by SILAC among genotypes (supplementary Table 5), as a further indication of the organelle dispersal in DKO cells. ARF1 was affected in clusters 170, 155, and 577 in fractions I, II and IV, respectively. Additional Golgi proteins, including GORASP2 (GRASP55), were changed in abundance in cluster 577 of fraction IV (supplementary Table 5). The steady state levels of GM130 (GOLGA2) were

unaffected in the post-nuclear supernatant proteomics but were perturbed in the fraction III analysed by SILAC (supplementary Tables 4 and 5).”

- *Page 10, “~ increased cholesterol esterification maintains low levels of cholesterol in the ER” the author already stated that there was no change in unesterified cholesterol in the previous paragraph. There is a conflict of statements.*

Unesterified cholesterol measured in lipidomics refers to all cholesterol found in cell membranes, most of which is located on the plasma membrane. In the text, we refer to cholesterol in the ER.

- *it would be great if any secretory vesicle proteins (biochemical data) are restricted to the fragmented Golgi compared to the normal ER or Golgi network. One possibility would be reduction of GRAP55 /65 protein since it is known with the fragmented Golgi (<https://doi.org/10.1186/s13024-022-00568-2>). Also, this phenomenon may be related to the human diseases HSP, ALS or other protein depot accumulation phenomenon. It would be better to discuss the possible relationship between ER tubule collapses, Golgi morphology changes in Erlins DKO associated with diseases if anything known.*

We have not directly checked if secretory vesicles proteins are restricted to the fragmented Golgi, however we checked the distribution of Golgi proteins in different fractions analysed by SILAC, and found that GRASP55 (also known as GORASP2) is one of the proteins that changes in cluster 577 in the SILAC analysis. We have now added a sentence to comment on these data (page 8, last paragraph, see above).

- *Also, it may leave a question if those fragmented Golgi is associated with lysosomes/peroxisome. Have you checked the lysosome/peroxisome structure as a control or colocalization?*

This is an interesting possibility, but we believe that it is outside the scope of the current manuscript, and should be explored in follow-up studies.

- *The mutant of G48V in Erlin2 is very interesting since any mutation construct containing G48V shows disappearance of the bands, probably destabilization of the protein. However, it is curious why it was made in Erlin2 although the clinical mutation of G to V occurs in Erlin1 (Navarino, G et al. 2014). It is mentioned that G50V in Erlin1 is related to the SPG62. It is questionable if the authors have observed the collapsed ER tubules or Golgi fragmentation in these cells. It may be directly related to human diseases.*

We have mutated the corresponding residue in ERLIN2 to compare this mutation with others identified in ERLIN2. To address the question whether this specific mutation affects the morphology of the Golgi or ER tubules, we would need access to patients’ derived cells, which are difficult to get owing to the rare occurrence of ERLIN1 mutations. Since the mutant G48V is unstable upon transfection, we believe that this variant corresponds to a loss-of-function allele. Further studies will

investigate which of the phenotypes that we describe here are also present in the individual ERLIN1 or ERLIN2 knock-out.

Minor changes

- *It seems better to follow and understand if the figure 2i, and 2j goes to Fig 3 at first since the paragraph in the results goes with these together.*

Thank you for this suggestion. We have changed the figures 2 and 3 accordingly.

- *reference page 26, Jo et al are same papers. Needs to be corrected.*

We have corrected this mistake.

- *Material Methods, correction period to comma for page 19*

*o 11.200 x g -> 11,200 x g
112 x g : is this really 112 x g or typo?
132.000 x g -> 132,000 x g*

We have carefully corrected and checked the material and methods. 112 x g is correct.

- *What are these abbreviations stand for GOMF , FFAT, VAPA ? They -latter two of them- look like directly from the reference.*

We have specified these abbreviations in the text and figure legends.

- *Page 9, the past paragraph needs a reference for "Esterification of excess cholesterol at the ER and its deposition in LDs are observed when cholesterol transport from the ER to the Golgi is impaired". Is this using ER to Golgi transport inhibited by a specific inhibitor?*

We added a reference to this paragraph.

Reviewer #2 (Comments to the Authors (Required)):

We thank the Reviewer for the positive comments.

I have only minor comments:

1. p. 10, top paragraph, referring to Fig. 8E. The authors state that avasimibe treatment reverted the effect of ERLIN DKO on the HMGCS mRNA. However, when looking at panel 8E, the avasimibe-induced elevation is still there, even though the expression levels are markedly lower than in the DMSO control. Please comment and rewrite.

We understand the point of the Reviewer, and we have rewritten this sentence to better reflect our results. Indeed, the difference in the transcriptional upregulation of

HMGCS is not statistically different anymore between DKO and WT upon treatment with avasimibe, but there is still a tendency for upregulation. The sentence now reads:

“SOAT1 inhibition suppressed the transcriptional upregulation of 3-hydroxy-3-methylglutaryl-CoA synthase (HMGCS), a SREBP target gene in both WT and DKO cells, blunting the difference between genotypes (Fig. 8E).”

2. p. 11, end of 2nd paragraph: '...however our data strongly argue against a direct interaction.' Please specify which data strongly argue against this.

This has now been rephrased, as follows: “FAF2 was also previously shown to interact with ERLIN2 upon overexpression (Wolf et al, 2021), and our data suggest that this interaction is mediated by TMUB1-L.”

In fact, our interactome analysis of TMUB1-L shows that FAF2 is significantly enriched. Together with the AlphaFold multimer simulations predicting interactions between TMUB1 and the ERLINs and TMUB1 and FAF2, these data suggest that the interaction of ERLIN with FAF2 is mediated by TMUB1. However, this is just a prediction and we have no experimental data to support it. Therefore, we have removed FAF2 and VCP from the model in the main figures (now in Figure 3A) and added a cartoon showing the predicted interaction of TMUB1 with FAF2 and VCP in the figure S2D.

3. p. 12, middle paragraph: 'Third, the presence of Arf1 and Arf4...'. This statement is relatively weak since the Arfs have not been primarily described as membrane contact site components. They were initially identified as Golgi vesicle transport machinery and later on in many other functions such as mRNA transport, mTORC1 activity and mitochondrial dynamics. The only Arf function associated with the present topic is that of Arf1 OSBP binding on trans-Golgi membranes. Please rewrite.

Thank you for this comment. We have removed the reference to Arf4 in this context.

Typos:

1. p. 5, line 2 from the bottom: '...and used it AS a control....'

2. p. 11, line 11: '...of unknowN significance in...'

We have corrected these typos, and carefully checked the rest of the manuscript

May 3, 2024

RE: Life Science Alliance Manuscript #LSA-2024-02620-TR

Prof. Elena I Rugarli
University of Cologne
Institute for Genetics
Joseph-Stelzmann-Str. 26
Koeln 50931
Germany

Dear Dr. Rugarli,

Thank you for submitting your revised manuscript entitled "ERLINS bridge TMUB1 and RNF170, limit cholesterol esterification, and regulate the secretory pathway". We would be happy to publish your paper in Life Science Alliance pending final revisions necessary to meet our formatting guidelines.

- please be sure that the authorship listing and order is correct
- please make sure the author order and names in your manuscript and our system match
- please add Keywords for your manuscript in our system
- please use the [10 author names, et al.] format in your references (i.e. limit the author names to the first 10)
- please add the Twitter handle of your host institute/organization as well as your own or/and one of the authors in our system
- please add callouts for Figure 5B and S5A

FIGURE CHECKS

- since Figure S4 only has one panel, please remove the "A" label from the figure and the legend

A. FINAL FILES:

B. MANUSCRIPT ORGANIZATION AND FORMATTING:

Sincerely,

May 7, 2024

RE: Life Science Alliance Manuscript #LSA-2024-02620-TRR

Prof. Elena I Rugarli
University of Cologne
Institute for Genetics
Joseph-Stelzmann-Str. 26
Koeln 50931
Germany

Dear Dr. Rugarli,

Thank you for submitting your Research Article entitled "ERLINS bridge TMUB1 and RNF170, limit cholesterol esterification, and regulate the secretory pathway". It is a pleasure to let you know that your manuscript is now accepted for publication in Life Science Alliance. Congratulations on this interesting work.

DISTRIBUTION OF MATERIALS:

Again, congratulations on a very nice paper. I hope you found the review process to be constructive and are pleased with how the manuscript was handled editorially. We look forward to future exciting submissions from your lab.

Sincerely,
